# Estimation of seasonal methane fluxes over a Mediterranean rice paddy area using the Radon Tracer Method (RTM)

Roger Curcoll[1], Alba Àgueda[3], Josep-Anton Morguí[2], Lídia Cañas[2], Sílvia Borràs[4], Arturo Vargas[1], Claudia Grossi[1,5]

[1] Institute of Energy Technologies, Universitat Politècnica de Catalunya, Barcelona, 08028, Spain
[2] Department of Evolutionary Biology, Ecology and Environmental Sciences, Universitat de Barcelona, Barcelona, 08028, Spain
[3] CERTEC, Chemical Engineering Department, Universitat Politècnica de Catalunya, Barcelona, 08019, Spain
[4] AIRLAB, Climate and Health Group, Barcelona Institute for Global Health (ISGlobal), Barcelona, Spain
[5] Physics Department, Universitat Politècnica de Catalunya, Barcelona, 08028, Spain

*Correspondence to*: Roger Curcoll (roger.curcoll@upc.edu)

**Abstract.** The Ebro River Delta, in the northwestern Mediterranean basin, has an extension of 320 km$^2$ and is mainly covered by rice fields. In the framework of the ClimaDat project, the greenhouse gases atmospheric station DEC was designed and installed in this area in 2013. The DEC station was equipped, among others, with a Picarro G2301 instrument and an ARMON (Atmospheric Radon Monitor) to measure both $CH_4$ and $CO_2$, and $^{222}Rn$ concentrations, respectively.

The variability of methane fluxes over this area and during the distinct phases of the rice production cycle was evaluated in this study by using the Radon Tracer Method (RTM). The RTM was carried out using: i) nocturnal hourly atmospheric measurements of $CH_4$ and $^{222}Rn$ between 2013 and 2019; and ii) FLEXPART-WRF back trajectories coupled with radon flux maps for Europe with a resolution of 0.05º x 0.05º available thanks to the project traceRadon. Prior to the calculation of methane fluxes by RTM, the FLEXPART-WRF model and the traceRadon flux maps were evaluated by modelling atmospheric radon concentrations at DEC station and comparing them with observed data.

RTM based methane fluxes show a strong seasonality with maximums in October (13.9 mg $CH_4$ m$^{-2}$ h$^{-1}$), corresponding with the period of harvest and straw incorporation in rice crop fields, and minimums between March and June (0.2 mg $CH_4$ m$^{-2}$ h$^{-1}$ to 0.6 mg $CH_4$ m$^{-2}$ h$^{-1}$). The total estimated methane annual emission was about 262.8 kg $CH_4$ ha$^{-1}$. These fluxes were compared with fluxes directly measured with static accumulation chambers by other researchers in the same area. Results show a stunning agreement between both methodologies, both having a similar annual cycle and monthly mean absolute values.

## 1 Introduction

Globally averaged surface $CH_4$ concentrations have risen from $722 \pm 25$ ppb in 1750 to $1927 \pm 2$ ppb in 2023, and in the last years (2020-2023), the global methane concentration has increased an average of 15 ppb year$^{-1}$ (Lan et al., 2024). The causes

of this increase are varied and still with large uncertainties (Drinkwater et al., 2023). The main driver of methane trend over the last decades is known to be the anthropogenic activity (Skeie et al., 2023), such as agriculture, fossil fuels combustion, and decomposition of landfill waste. In addition to the direct methane emissions into the atmosphere, methane increase is also driven by CO and $NO_x$ emissions, which change the atmospheric oxidation capacity and hence atmospheric methane lifetime (Wuebbles and Hayhoe, 2002). A reduction of all anthropogenic methane sources is therefore mandatory to reduce the increase in concentrations and reach the Paris agreements (Schleussner et al., 2016).

Particularly, in the case of agriculture, it is known that over the past 110 years global $CH_4$ emissions from rice cultivation have increased by 85% due to rice field expansion and nitrogen fertilizers use (Zhang et al., 2016). Global rice fields are estimated to emit between $18.3 \pm 0.1$ Tg $CH_4$/yr and $38.8 \pm 1.0$ Tg $CH_4$/yr, with emissions varying based on different water management practices (Yan et al., 2009; Zhang et al., 2016). Rice field methane emissions follow a strong seasonality mainly due to the management practices. Flooded rice paddies and wetland environments have a predominantly oxygen-free (anoxic) soil profile. In these ecosystems, $CH_4$ is produced by methanogenic bacteria that digest organic matter under anaerobic conditions (methanogenesis) (Zhang et al., 2016). Atmospheric $CH_4$ concentrations measured in the lower boundary layer of these ecosystems result from a combination of processes, including diffusion, ebullition and transport through aerenchyma of the plants. This methane originates from the net $CH_4$ produced at the soil-water/soil-atmosphere interface of the ecosystem, further influenced by both positive or negative contributions due to the atmospheric mixing and advective transport from remote areas.

So far, many studies have investigated the different factors and variables controlling methane emissions from rice paddies, including both environmental and agricultural considerations. As an example, it has been observed that during the crop cycle these factors may include soil and air temperature, soil redox potential, water management, organic amendment or fertilizers management (Oo et al., 2015; Pereira et al., 2013; Sass et al., 1991; Seiler et al., 1983; Wang et al., 2018; Yan et al., 2005). In recent years, some efforts have been done to monitor also $CH_4$ emissions during fallow periods of rice soils. This includes investigations into the impact of straw management practices (e.g. incorporation into the field, removal from the field, or burning) and flooding practices after harvest, as these can substantially influence emission levels (Alberto et al., 2015; Martínez-Eixarch et al., 2018; Fitzgerald et al., 2000; Belenguer-Manzanedo et al., 2022).

The results of these studies may be of great utility to understand the emission differences due to diverse agricultural practices and soil characteristics, thus offering valuable insights for improving agricultural techniques and protocols. In addition, such studies are needed to improve emission inventories as well as methane emission models.

Nowadays various approaches have been applied to estimate $CH_4$ emissions from rice fields. These approaches include direct flux measurements using techniques such as the eddy-covariance method (e.g. Alberto et al., 2015; Iwata et al., 2018; Runkle

et al., 2019), accumulation chambers (Martínez-Eixarch et al., 2021; Wassmann et al., 2000), or measuring methane both below and above the canopy (Simpson et al., 1995). A combination of all these techniques (Meijide et al., 2011) has also been valuable to provide a comprehensive understanding of $CH_4$ emissions. Top-down techniques have also been used to estimate methane fluxes on rice fields, such as aircraft measurements (Desjardins et al., 2018; Peischl et al., 2012) or inversion models from atmospheric measurements (Thompson et al., 2015) or satellite data (Chen et al., 2022). However, in studies where several approaches are used, some disagreement have been found, mainly due to the uncertainties associated with atmospheric transport models or the accuracy of the emissions inventories (Desjardins et al., 2018; Cheewaphongphan et al., 2019).

One of these previous methods is the one know as Radon Tracer Method (RTM). The RTM has been used in different sites for the retrieval of fluxes of greenhouse gases (GHG) and other trace gases (Grossi et al., 2018; Levin et al., 2011; Schmidt et al., 1996; Vogel et al., 2012). The RTM uses co-located atmospheric observations of the noble gas $^{222}$Rn and the gas of interest, in this case $CH_4$, together with modelled values of $^{222}$Rn fluxes. The utility of this method has been confirmed in the recent years and the Integrated Carbon Observation System (ICOS) is currently including atmospheric radon measurements within its network for offering GHG fluxes based on RTM applications. Actually, an interactive tool to apply the RTM to estimate GHG fluxes from ICOS atmospheric concentration measurement was developed as reported by Yver-Kwok et al. (2024). However, because a harmonized protocol for the RTM application is not yet available, researchers are now focusing on evaluating RTM limitations to improve its application worldwide (Levin et al., 2021; Yver-Kwok et al., 2024).

In the present work, methane fluxes over a rice paddies area, located in the Ebro River Delta, were estimated during different phases of the rice cultivation cycle. The estimation was conducted using the RTM, which was applied in a frontier region such the Ebro River Delta, and a validation of the methodology was also performed.

The present work presents the area of study and the applied methodology in the *Methods* section. In the *Results and discussion* section the hourly radon and methane atmospheric measurements are firstly presented, together with: i) hourly modelled atmospheric radon concentrations; ii) RTM based methane fluxes. The reliability of the radon flux maps and transport models used for the area and period of interest was also evaluated and is presented in this section. Finally, $CH_4$ fluxes obtained with the application of the RTM were compared with fluxes from known emission inventories (i.e. EDGAR) and previous research studies based on different methodologies.

## 2 Methods

Here the full methodology and the different steps designed and realized for the calculation of methane fluxes over the Ebro River Delta is presented. Figure 1 shows a scheme of the different inputs/outputs participating into this process as it will be explained in detail in the following subsections.

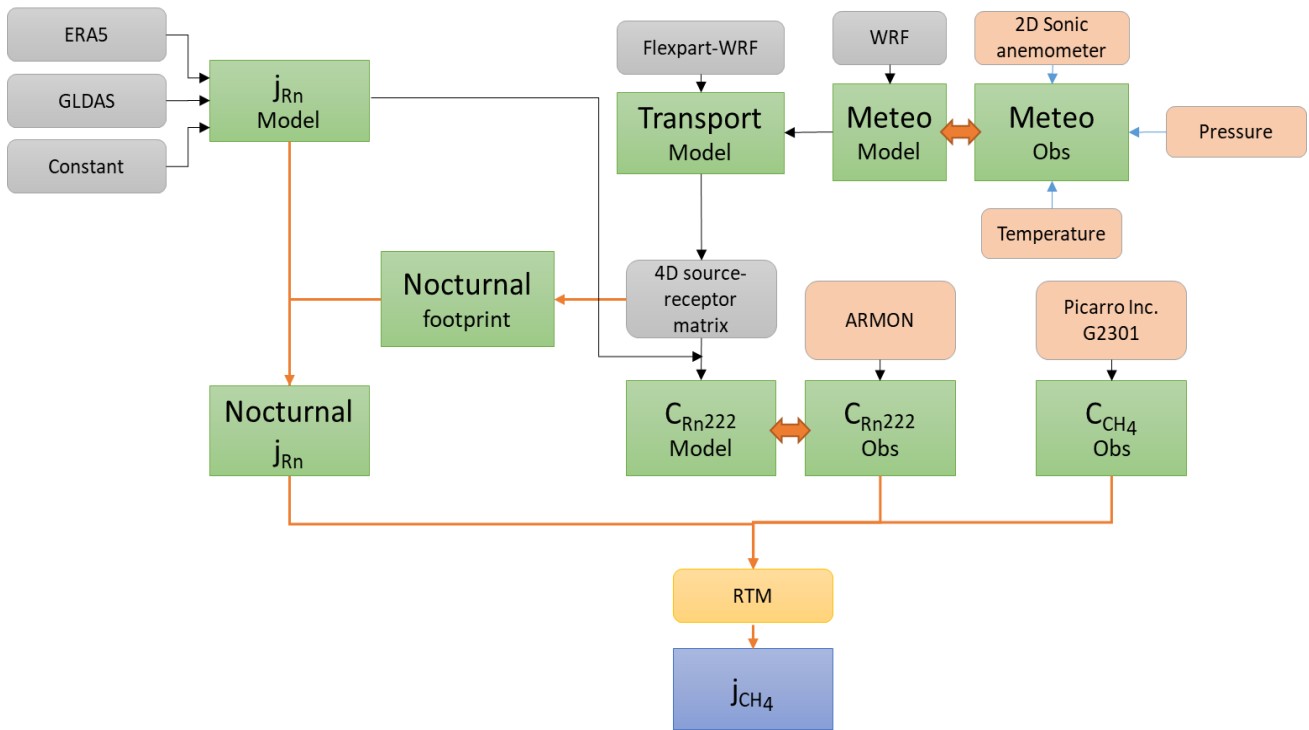

90

**Figure 1. Sketch of the process followed for estimating CH₄ fluxes ($j_{CH_4}$) in this study.**

## 2.1 Site description: Ebro River Delta

95    The Ebro River Delta (ERD), with an extension of 320 km², is located at the Ebro River mouth, in the Spanish coast of the western Mediterranean basin. Its main land use is rice field (70 %), followed by beaches, salt marshes, dunes and coastal lagoons, according to the CORINE land use inventory (European Union, 2018) (See Supplement, Figure S1).

The ERD experiences strong winds coming from the North of Spain and channeled through the Ebro River watershed (Gangoiti et al., 2002; Valdenebro et al., 2011). These winds cross the valley between the Iberian System and the Pyrenees. The wind

regime in the ERD is also dominated by land-sea breeze phenomena, with upcoming winds from the sea during the day and land-sea breezes at night (Martín et al., 1991).

The ERD has a typical Mediterranean climate with mild winters and warm summers (Casanova, 1998). Wind blows with high mean annual velocities ($> 8$ m s$^{-1}$) during the entire year (Generalitat de Catalunya, 2022), and blows predominantly from the NW in winter (e.g. Casanova, 1998), and from the S-SE in summer (Generalitat de Catalunya, 2022). The atmospheric relative humidity is high over the entire year ($> 65\%$) (e.g. Grossi et al., 2016).

The ERD has a flat orography, with approximately 60% of its total area having elevations lower than 1 m above sea level (a.s.l.) (Generalitat de Catalunya, 2022). Two main canals flank the river, distributing water across a network of smaller canals. Rice paddies cover an extension of more than 200 km$^2$ and represent an 83% of the total crop area in the Ebro Delta.

Figure 2 presents a Gantt diagram, adapted from Àgueda et al. (2017), outlining the main anthropogenic activities conducted in the ERD rice fields. It is important to note that the timing of these activities may vary slightly from year to year due to the weather seasonality or changes in management practices.

Rice fields in the ERD remain completely flooded during the majority of the growth cycle, with a water column typically ranging from 8 cm to 15 cm (Alvarado-Aguilar et al., 2000). Prior to irrigation, usually in mid-April, the land is prepared (tilled and leveled) and fertilized. After the flooding, direct sowing takes place between mid-April and mid-May and plants grow until mid-August, marking the onset of harvesting. After harvest, rice straw is incorporated into the soil using mechanical means. In some years, a flooding with sea water of some of the rice fields was carried out during winter months in order to cope with an ampullariidae plague.

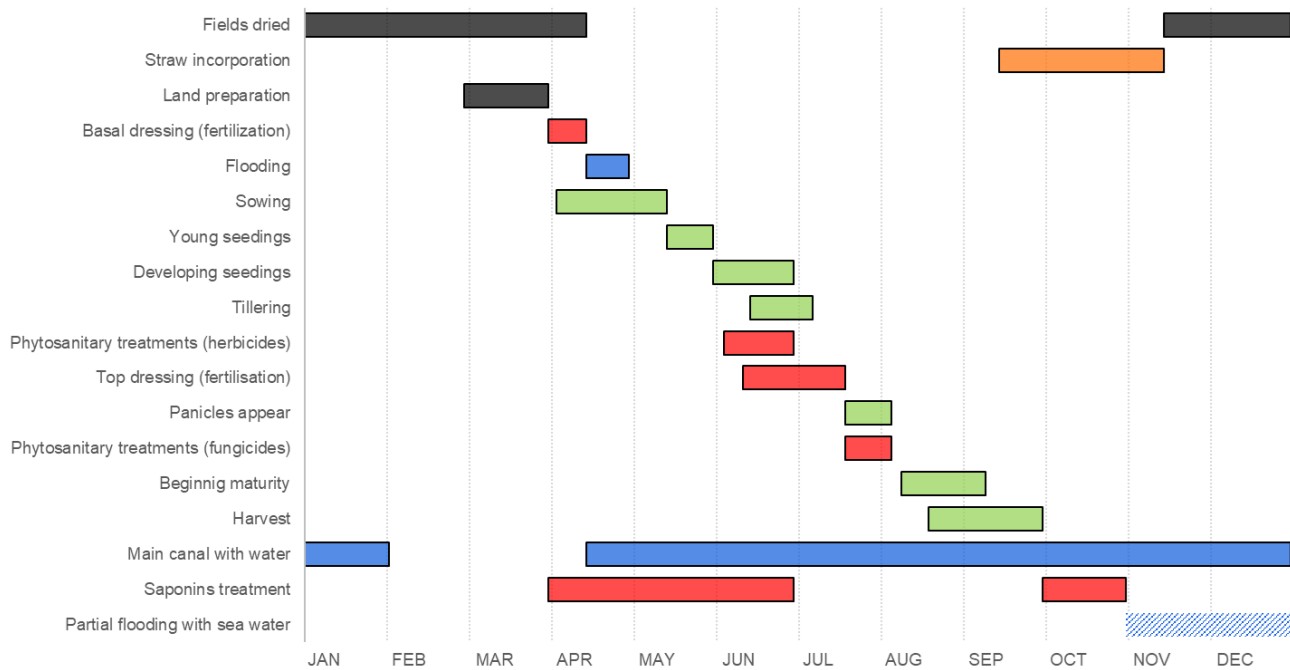

**Figure 2. Gantt chart of the annual agricultural practices usually performed in the Ebro River Delta (ERD) (black: fields; orange: straw and weed management; red: chemicals; blue: water management; green: rice phenology). Adapted from Àgueda et al. (2017).**

## 2.2 Atmospheric observations

An atmospheric station in the Ebro Delta (DEC, 40.74N; 0.79E, 7 m a.s.l.) was built in 2013 within the ClimaDat project (Grossi et al., 2016; Morgui et al., 2013). The station was built next to the Biological Station of Canal Vell, in the middle of the ERD, surrounded by rice fields (black asterisk in the right panel of Figure S1 of Supplement). At the DEC atmospheric station, GHG ($CO_2$, $CH_4$, $N_2O$, CO), atmospheric radon and meteorological variables (see section 2.2.4) were continuously measured at a 10 m above ground level (a.g.l.) tower. Due to the extreme humidity conditions and salty air at DEC site, sampling pumps continuously broke over the years and thus only 30% of the days in the sampling period (2013-2019) have the full record of $^{222}$Rn, GHG, and meteorology variables.

## 2.2.1 Atmospheric radon measurements

The atmospheric concentrations of the radioactive and noble gas radon ($^{222}$Rn) were hourly measured at DEC station using a first version of the Atmospheric Radon Monitor (ARMON) designed and calibrated by researchers of the IONHE (Ionizing Radiation, Health and Environment) group of the Institute of Energy Techniques (INTE) of the Universitat Politècnica de Catalunya (UPC, Spain). The ARMON is based on the alpha spectrometry of positive ions of $^{218}$Po, coming from the radon decay within the detection volume, collected on a Passivated Implanted Planar Silicon (PIPS) detector surface by an electrostatic field (Grossi et al., 2012; Vargas et al., 2015). The ARMON is capable to distinguish between $^{222}$Rn and $^{220}$Rn (thoron) contribution and, with an integration time of 1 hour, has a detection limit of 0.132 Bq m$^{-3}$ and a total uncertainty around 10% for average concentrations of about 5 Bq m$^{-3}$ (Curcoll et al., 2024; Röttger et al., 2025). This type of monitor was installed at several Spanish stations (Grossi et al., 2016) and its response and performance have been compared with those of other radon and radon progeny monitor types ( Grossi et al., 2016; Grossi et al., 2020; Röttger et al., 2025).

Due to the fact that the collection efficiency of $^{218}$Po on the detector surface is strongly influenced by the humidity of the sampled air, a low-maintenance drying system was designed and installed at the DEC site (see section 2.2.3). Moreover, to address this influence, a linear water correction factor was empirically determined and applied following the methodology outlined by Grossi et al. (2012).

## 2.2.2 Atmospheric CH$_4$ measurements

CH$_4$ measurements were continuously performed by a G2301 gas concentration analyzer (Picarro Inc., USA). This device is based on the cavity ring-down spectroscopy technique (CRDS) (Crosson, 2008) and offers simultaneous and precise measurements of CO$_2$, CH$_4$ and H$_2$O every 5 seconds. Hourly mean values were used in this study.

During the measurements period the Picarro G2301 analyzer was calibrated every 2 weeks using four secondary working gas standards, which were calibrated at the beginning and at the end of their lifetime against seven standards of the National Oceanic and Atmospheric Administration (NOAA). Calibration scales were WMO-X2019 (Hall et al., 2021) and WMO-X2004A (Dlugokencky, 2005) for CO$_2$ and CH$_4$, respectively. A fifth target gas was analyzed daily for 20 min in order to check the stability and quality of the instrument calibration. Precision of the instrument for methane was better than ±0.3 ppb and accuracy better than ±1 ppb.

Although the instrument at DEC site was measuring dried air, a water correction factor was applied for a better accuracy of the measurements following the MPI-Jena method (Rella et al., 2013).

### 2.2.3 Drying system

As previously noted, water vapor content has an important influence in Picarro Inc. measurements (Rella et al., 2013; Reum et al., 2017) as well as radon measurements with ARMON (Grossi et al., 2012; Curcoll et al. 2023). Moreover, the extreme weather conditions at the ERD during the summer season, characterized by temperatures surpassing 30 ºC and relative humidity levels reaching 80%, highlights the need for sample drying to prevent water condensation in the pipes or the instruments. To address this concern, an automatic circuit was developed at DEC station to dry the air sample before it entered the instruments. The sampled air (2.5 L min$^{-1}$) was passed through a Nafion® membrane (Permapure, PD-100T-24MPS) exchanging water molecules with a dry counter-current air flow. The counter-current air flow was generated in a two-step process, first flushing air through a cooling coil in a refrigerator at 3 °C and a pressure of 5.5 barg, and then through a cryotrap at -70 °C and a pressure of 1.5 bar. Multiple cryotraps were selected with electrovalves to increase the autonomy of the system to approximately 2 months. After the Nafion membrane, the air sample had a water vapor concentration between 100 ppm and 400 ppm. At this point, the flow was divided: 2 L min$^{-1}$ were sent directly to the ARMON and the rest were passed through a cryotrap in order to reduce the water content up to 10 ppm for the Picarro Inc. G2301 instrument.

### 2.2.4 Meteorological observations

Meteorological variables were continuously measured at the DEC tower. The tower was equipped with: (1) a two-dimensional sonic anemometer (WindSonic, Gill Instruments) for wind speed and direction (accuracies of ± 2% and ± 3 degrees, respectively); (2) a humidity and temperature probe (HMP 110, Vaisala) with an accuracy of ± 1.7% and ± 0.2 °C, respectively; (3) a barometric pressure sensor (61302V, Young Company) with an accuracy of 0.2 hPa (at 25 °C) and 0.3 hPa (from -40 °C to +60 °C). All the accuracy factors previously mentioned refer to manufacturers' specifications.

### 2.3 CH$_4$ fluxes estimation using the Radon Tracer Method (RTM)

The Radon Tracer Method (RTM) was applied in this work to obtain nocturnal methane fluxes [mg CH$_4$ m$^{-2}$ h$^{-1}$] over the footprint area covered by the DEC station. The RTM uses atmospheric concentration measurements of $^{222}$Rn [Bq m$^{-3}$] and the target gas (here, CH$_4$ [mg CH$_4$ m$^{-3}$]) together with simulated values of $^{222}$Rn fluxes [Bq m$^{-2}$ s$^{-1}$]. This method, described in detail in the works from Grossi et al. (2018), Levin et al. (2011, 2021), Schmidt et al. (1996) or Vogel et al. (2012), is based on the assumption that the nocturnal lower atmospheric boundary layer can be described as a well-mixed box of air. The nocturnal boundary layer effective height ($h(t)$) is considered homogeneous within the box and horizontal advection is considered negligible under stable atmospheric conditions (Griffiths et al., 2013). Thus, within this atmospheric volume the variation of the concentration of any tracer (represented with the subindex i) with time $\frac{dc_i(t)}{dt}$ is proportional to the flux of the tracer itself $j_i(t)$, inversely proportional to the height $h_i(t)$, and homogenous within the volume (see Eq. 1).

$$\frac{dC_i(t)}{dt} \propto \frac{j_i(t)}{h_i(t)} \tag{1}$$

In the case of $^{222}$Rn we should also consider its decay by including a decay constant ($\lambda_{Rn}$; [s$^{-1}$]) (see Levin et al., 2021).

If the RTM methodology is applied for single nocturnal windows, the gases fluxes may be taken as constant within each individual nocturnal window and finite temporal concentration increases (or slopes) of the measured gases may be used. Finally, as both gases are measured at the same point, the effective height $h(t)$ may be the same for both. Combining Eq. 1 for the measured target gas (CH$_4$) as well as for $^{222}$Rn, the term $h(t)$ can be removed, obtaining Eq. 2, where the target gas flux $j_{CH_4}$ can be calculated.

$$j_{CH_4} = j_{Rn} \frac{\Delta C_{CH_4}(t)}{\Delta C_{Rn}(t)} \left(1 + \frac{\lambda_{Rn} \cdot C_{Rn}(t)}{\Delta C_{Rn}(t)/\Delta t}\right)^{-1} \tag{2}$$

In Eq. 2, for each night, $j_{Rn}$ is the radon flux, $\Delta C_{Rn}$ is the radon atmospheric variability over the nocturnal window, and $\Delta C_{CH_4}$ is the methane atmospheric variability over the same time interval. Considering that applying the RTM during the nocturnal window the maximum change in $^{222}$Rn activity concentration due to radioactive decay is less than 10%, which is much smaller than the uncertainties due to RTM and radon exhalation maps, the decay contribution of radon may be neglected (Levin et al., 2021), obtaining the simplified Eq. 3.

$$j_{CH_4} = j_{Rn} \frac{\Delta C_{CH_4}}{\Delta C_{Rn}} \tag{3}$$

Thus, from Eq. 3, if the radon flux over the footprint area is known, the methane flux can be calculated knowing the temporal variation of radon and methane atmospheric concentrations measured during each nocturnal window.

To estimate the effective nocturnal radon flux over the footprint area (i.e. around DEC station), according to the methodology presented by Grossi et al. (2018), a window of 70 km x 70 km around it was selected as feasible influence area. The influence area for radon flux retrieval for every single night over the whole 2013-2019 period was calculated from the residence time of 6 h FLEXPART-WRF back trajectories from the DEC station. The setup of both WRF and FLEXPART models is described in section 2.4.2. Representative back trajectories were run daily at 00h UTC, and only the layers next to the surface (0 m-200 m) and within the 70 km x 70 km window were considered. Three average radon flux values for every nocturnal event (denoted as $j_{Rn}$ in Eq. 3) were derived by multiplying daily footprints with three different European radon exhalation maps (refer to section 2.4.1),

As the RTM is based on the stability assumption, only night periods with specific characteristics were chosen in this study. The selection criteria were based on the following requirements:

 i) A nocturnal window between 21h UTC and 03h UTC was selected for each single night analysis in order to use only nocturnal accumulation events.

ii) A data selection criterion based on a threshold of $R^2 \geq 0.5$ for the linear correlation between $^{222}$Rn and $CH_4$ concentrations
was used to reject events with a low linear correlation between the atmospheric concentrations of both gases.

 iii) Only nights where both $CH_4$ and $^{222}$Rn had a positive concentration gradient were selected, in order to retrieve only positive net fluxes under stable boundary layer conditions.

iv) In order to evaluate the possible effect of advection signals for both radon and methane accumulations, the RTM was run both considering only events with wind speeds below 1.5 m s$^{-1}$ and without wind speeds restriction.

**2.4 Evaluation of the reliability of RTM-based CH₄ fluxes**

As explained in the previous section, the RTM is based on some assumptions, but no harmonized protocols are so far available for its applications. One of the main questions that arises when applying the RTM methodology is the representativeness area of the estimated fluxes. Levin et al. (2021) exposed that one of the main limitations of the RTM was that the quantitative comparison of RTM-based with bottom-up emission data was not directly possible without reliable footprint modelling of the
night-time observations, and that this may be hampered by the reliability of the transport model under nocturnal conditions (i.e. boundary layer height, wind speeds, etc.).

Before applying the RTM for the calculation of methane fluxes over the ERD area, the effective nocturnal radon flux term (denoted as $j_{Rn}$ in Eq. 3) seen by the DEC station each night was estimated, in agreement with Grossi et al. (2018), using together the meteorological (WRF) and transport (FLEXPART-WRF) models. Additionally, as previously mentioned, three
different radon exhalation maps were used to assess the results obtained for each of them (see Fig. 1).

To evaluate the performance of the FLEXPART-WRF and of the three radon flux maps (ERA5-Land, GLDAS-Noah and constant map, as explained later in subsection 2.4.1) atmospheric radon concentrations at DEC were simulated for a whole year (2019). Simulated hourly radon concentration were obtained by using the output of the FLEXPART-WRF model to build a source-receptor matrix (Seibert and Frank, 2004) which was then coupled with one of the three different available radon

exhalation maps. Modelled hourly radon concentrations were then compared with observed data measured by ARMON. Details of this procedure are explained in detail in the following sections.

### 2.4.1 Radon exhalation maps

Radon exhalation maps used in this study were obtained from the European radon maps developed by Karstens and Levin (2023) within the EMPIR 19ENV01 traceRadon project (Röttger et al., 2021). The theoretical equations applied to simulate
the radon transport in the soil and its exhalation to the lower atmosphere are described in Karstens et al. (2015). It basically assumes that the transport of radon through the soil and across the soil surface into the atmosphere occurs predominantly by molecular diffusion and it strongly depends on physical soil parameters and its water content (Nazaroff, 1992). The model uses soil uranium content (Cinelli et al., 2019), soil properties (Hiederer, 2013) and two different soil moisture reanalysis datasets: ERA5-Land soil moisture reanalysis (Muñoz Sabater, 2019) or GLDAS-Noah v2.1 soil moisture reanalysis
(Beaudoing and Rodell, 2020). For this study, monthly data were used for the period 2013-2016, and daily data were used for the period 2017-2019, in accordance with model output availability. The horizontal resolution of these radon exhalation maps is 0.05º x 0.05º. Two radon exhalation maps were obtained using both ERA5-Land (ERA5) and GLDAS-Noah (GLDAS) datasets. In addition, a third radon exhalation map (Constant) was generated with a constant term exhalation from inland surface grid cells of 15.8 mBq m$^{-2}$ s$^{-1}$ and a zero radon exhalation flux for sea grid cells. These values were applied according
to previous European studies (Arnold, 2009; Levin et al., 1999; Schmidt et al., 2001). Figure S2 of the Supplementary material shows average radon exhalation distribution for the period 2013-2019 for Europe and for the ERD area according to the three previous maps.

### 2.4.2 WRF and FLEXPART-WRF simulations for RTM

The Lagrangian particle dispersion model FLEXPART-WRF v.3.1 (Brioude et al., 2013) was used to calculate back trajectories
from DEC station. The original FLEXPART model (Stohl et al., 2005) was designed for calculating long-range and mesoscale dispersion of hazardous substances from point sources, but evolved into a comprehensive tool for multi-scale atmospheric transport modelling and analysis (Pisso et al., 2019). In this work, we used the FLEXPART version that works with the inputs coming from the mesoscale meteorological model Weather Research and Forecasting (WRF, Skamarock et al., 2021). The decision of using a mesoscale model, such as WRF, for this study rather than a global model was made based on the dimensions
of the ERD (15 x 22 km$^2$) and the recognized importance of using high-resolution mesoscale models in coastal areas (Ahmadov et al., 2009; Hegarty et al., 2013). The FLEXPART-WRF v.3.1 model, referred to as Flex-WRF hereafter, has already been used in studies where global weather models may not reproduce correctly the terrain-induced weather features due to complex terrains (e.g. coastal sites or mountains) (Aliaga et al., 2021; Madala et al., 2016). The WRF model v.4.1 (Skamarock et al., 2021) was set up for this study with three domains (see Appendix): d01) Europe (spatial resolution of 27 km x 27 km); d02)

Iberian Peninsula (spatial resolution of 9 km x 9 km), and d03) Northwestern Spain (spatial resolution of 3 km x 3 km). All domains had 57 verticals layers up to 50 hPa and the meteorological initial and lateral boundary conditions were determined using ERA5 global model data (Hersbach et al., 2020). More details about the parametrization used for these simulations are shown in the Appendix.

WRF outputs were used as inputs within Flex-WRF model to simulate back trajectories arriving at DEC station inlet point (10 m a.g.l.). WRF outputs from all 3 domains were used. The back trajectories were run simulating the transport of 10,000 particles with time steps of 1 hour. The output of this type of back trajectory simulations is the residence time of the particles in each 3D grid cell at every time step (1h).

Flex-WRF back trajectories were used both to simulate the radon concentrations in DEC (see section 2.4.3) and to retrieve the effective radon flux influencing the DEC station each night for the RTM application. For the radon concentration simulation, back trajectories of 8 days (192h) were used. For the retrieval of the nocturnal effective radon flux, back trajectories length was set to 6 hours.

The output domain of Flex-WRF (see Figure A1) covered Europe and the north-Atlantic region with a resolution of 0.1º x 0.1º, although a nested output domain of 150 km x 150 km around DEC station with a resolution of 0.05º x 0.05º (referred as Flexpart NEST, Figure A1) was also used. The vertical resolution of the output was from 0 to 5,000 m height (17 levels). For the retrieval of nocturnal radon fluxes, only the nested domain was used.

From all the back trajectories, a 4D source-receptor matrix (Seibert and Frank, 2004) for particles arriving at DEC was obtained. A $^{222}$Rn decay ($t_{1/2}$ = 3.8 days) was applied to the matrix in order to obtain the source-receptor matrix for $^{222}$Rn. The layers with influence for the source-receptor matrix were assumed to be only those below 200 m (Hüser et al., 2017).

Figure S3 of the Supplementary material shows two examples of the residence time of the fictitious particles calculated with Flex-WRF for two of the most typical synoptic situations using 192 hours back trajectories.

### 2.4.3 Modelled hourly radon concentrations at DEC station

The 8 days Flex-WRF back trajectories were run at every hour for every day of 2019 in order to simulate hourly atmospheric radon concentrations at DEC for this year. Three radon concentration time series at DEC station were then simulated at every hour by multiplying the source-receptor matrix with each of the three different radon exhalation maps presented in section 2.4.1 and dividing them by the height of the influence layer (i.e. 200 m), obtaining: Flex-WRF-ERA5, Flex-WRF-GLDAS and

Flex-WRF-Const time series, respectively. The largest domain from the back trajectory simulations was rescaled to 0.05° x 0.05° and merged with the nested domain to have the same resolution as the radon exhalation maps.

### 2.4.4 Statistical metrics to evaluate Flex-WRF-based $^{222}$Rn concentrations.

For the quantitative evaluation of the goodness of the simulation of radon concentrations at DEC, the following metrics were
calculated between simulated and observed hourly radon concentrations in 2019: the bias (BIAS), the correlation coefficient (R), the root mean square error (RMSE) and the weighted root mean square error (WRMSE). This last coefficient was calculated as in Eq. 4, and the weight was defined as the average value between observed and modelled values (Eq. 5). The WRMSE can better evaluate the performance of the models without giving too much importance to nocturnal overestimations or underestimations of concentrations due to a poor representativeness of the local boundary layer height (Arnold, 2009).

$$WRMSE = \sqrt{\frac{1}{N}\sum_{i=1}^{N}\frac{(x_i^m - x_i^o)^2}{\tilde{x}_i^2}}$$    (4)

with

$$\tilde{x}_i = \frac{x_i^m + x_i^o}{2}$$    (5)

where $x_i^o$ refers to the measured values and $x_i^m$ to the modelled ones.

### 2.5 Literature review of CH$_4$ fluxes in the ERD area

To assess the reliability of the methodology applied in this work, methane flux values derived from the RTM were compared against data from available databases, such as the Emissions Database for Global Atmospheric Research database (EDGAR), as well as from Spanish inventories and experimental studies (Martínez-Eixarch et al., 2018, 2021).

EDGAR v.7.0 inventory, developed by the European Commission Joint Research Centre and the Netherlands Environmental Assessment Agency (European Commission, 2023), includes global anthropogenic emissions of GHGs and air pollutants by
country on a spatial grid. The EDGAR version used in the present study provides monthly CH$_4$ emissions on a 0.1° x 0.1° resolution for the period 2013-2019. All major anthropogenic source sectors (e.g., waste treatment, industrial and agricultural sources) are included in this inventory, whereas natural sources (e.g. wetlands or rivers) are excluded. The spatial allocation of emissions on 0.1° x 0.1° grid cells in EDGAR has been built up by using spatial proxy datasets with the location of energy and manufacturing facilities, road networks, shipping routes, human and animal population density, and agricultural land use.
Figure 3 shows the EDGAR inventory grid map extracted for a region centered over the ERD.

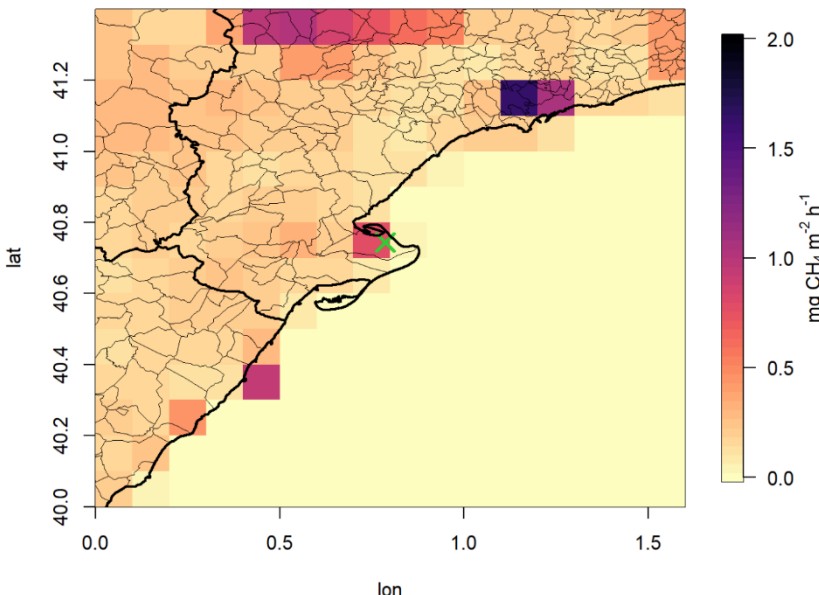

**Figure 3. Average methane fluxes around ERD station (150 km x 150 km) according to EDGAR inventory v.7.0 for the period 2013-2019. DEC station is indicated with a green cross.**

National inventory reports stablish, following the methodology of the IPPC Guidelines (Eggelston et al., 2006), an annual emission factor for rice crops as a function of a fixed term multiplied by a series of coefficients associated with fertilization management or length (in days per year) of rice crop production. The inventoried emissions also take into consideration the fallow emissions, although it distributes its emissions among the crop season. In the latest national inventory report of the UNFCC (National Inventory Report of Spain, 2023), Spain reported an average methane emission flux of 1.32 kg $CH_4$ ha$^{-1}$ day$^{-1}$ for its rice crops during crop cultivation period (150 days), equivalent to 5.54 mg $CH_4$ m$^{-2}$ h$^{-1}$ during the crop period and a total yearly emission of 198.5 kg $CH_4$ ha$^{-1}$. Following the IPCC methodology only for the Ebro Delta crop fields, the same value is obtained.

In 2015 and 2016 a multisite field experiment covering the agronomic and environmental variability of the rice growing area of the ERD area was conducted by researchers of the Institute of Agrifood Research and Technology (IRTA, Spain) (Martínez-Eixarch et al., 2018, 2021) for evaluating the GHG emissions during the productive (June-October) and fallout (October-December) rice seasons. Static flux chambers were used in this study at 24 sampling points, covering both sides of the river and different rice varieties and fertilization management practices present in the area. Annual methane emissions obtained from these studies were 262.6 ± 5.9 kg $CH_4$ ha$^{-1}$, equivalent to an average flux of 3.0 mg $CH_4$ m$^{-2}$ h$^{-1}$.

**3 Results and discussion**

**3.1 Observed atmospheric concentrations of $^{222}$Rn and CH$_4$ at DEC station (2013-2019)**

Figure 4 shows monthly average values of $^{222}$Rn and CH$_4$ atmospheric observations measured at DEC station during the period 2013-2019. Atmospheric CH$_4$ concentrations show a pronounced seasonal trend, with maximums observed in the months of September, October and November, with monthly average concentrations between 2.2 ppm and 2.4 ppm, and minimums from March to July with monthly average concentrations below 2 ppm. The highest methane concentrations correspond to months of the year during which straw incorporation occurs, as reported in Figure 2. Conversely, monthly averages of the atmospheric

radon concentrations do not show any strong seasonality. In general, monthly mean values are below 4 Bq m$^{-3}$, lower than those usually measured at continental sites (Grossi et al., 2016, 2018; Levin et al., 2021) but similar to those observed at coastal sites (Biraud et al., 2000; Vargas et al., 2015). Higher radon values are observed in December, as previously reported by Grossi et al. (2016) for the period 2013-2015 at the same station. This could be attributed to the arrival of northwestern winds from continental areas in the north of Spain to the DEC station, probably with air masses rich in radon in comparison with

background levels (see Grossi et al. 2016).

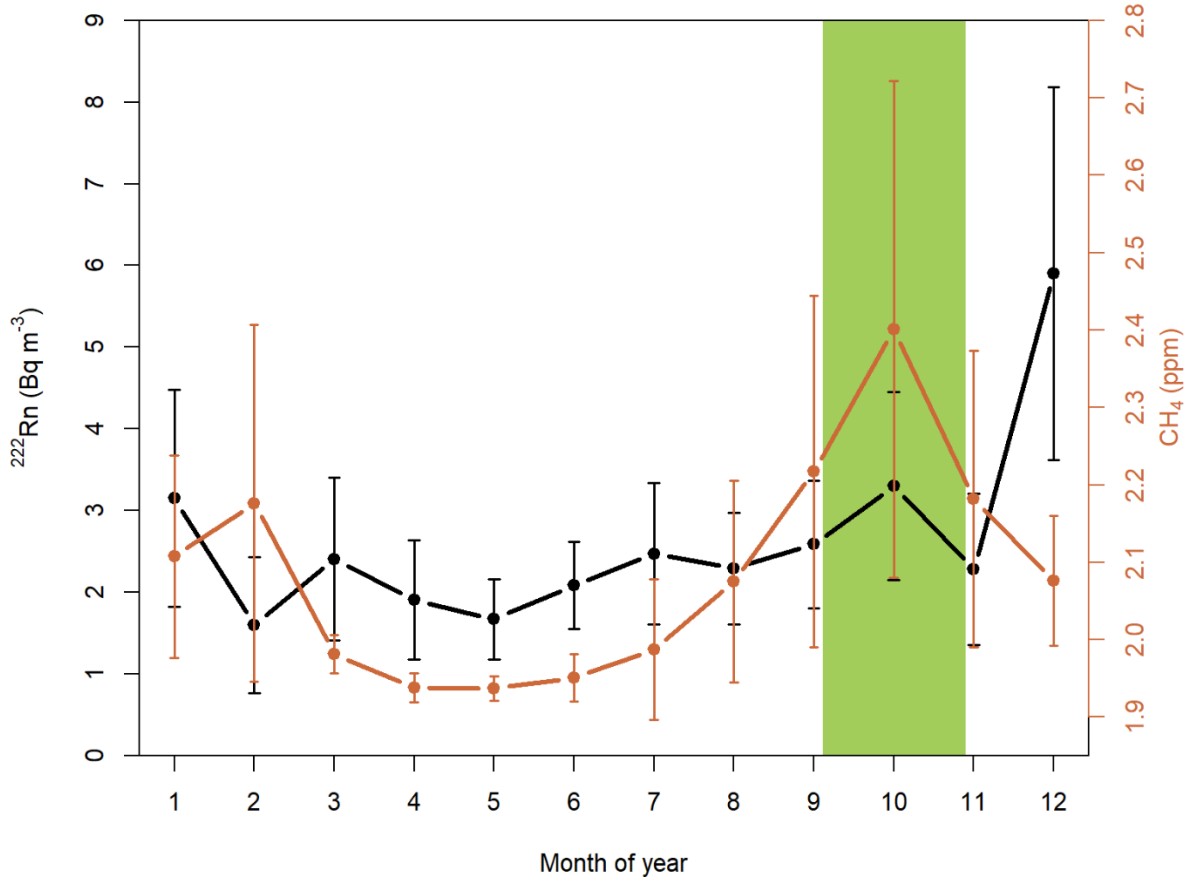

**Figure 4. Observed average annual cycle for radon (black) and CH₄ (orange) concentrations at DEC within the 2013-2019 period dataset. Vertical whiskers represent variability (standard deviation) for each month. Green area corresponds to the "straw incorporation" period in the rice management cycle at ERD.**

Average monthly diurnal cycles for both $CH_4$ and $^{222}Rn$ gases have been calculated for each month of the year over the whole 2013-2019 dataset (Figure 5). Methane concentrations show a flat diurnal cycle from December to July. However, from August to November a more prominent methane diurnal cycle can be observed, with the typical nocturnal accumulations and the decrease of concentrations after 06h UTC. This may indicate the accumulation, during nocturnal stable conditions, of local methane emissions. On the other hand, the hourly average radon concentrations show a more regular diurnal cycle throughout

the year, with a daily maximum at 07h UTC and minimums in the afternoon. These asymmetric differences in the cycles between the two gases cannot be only explained by atmospheric conditions and it could be due to seasonal differences in the source terms of the two gases as it will be analyzed in more detail later.

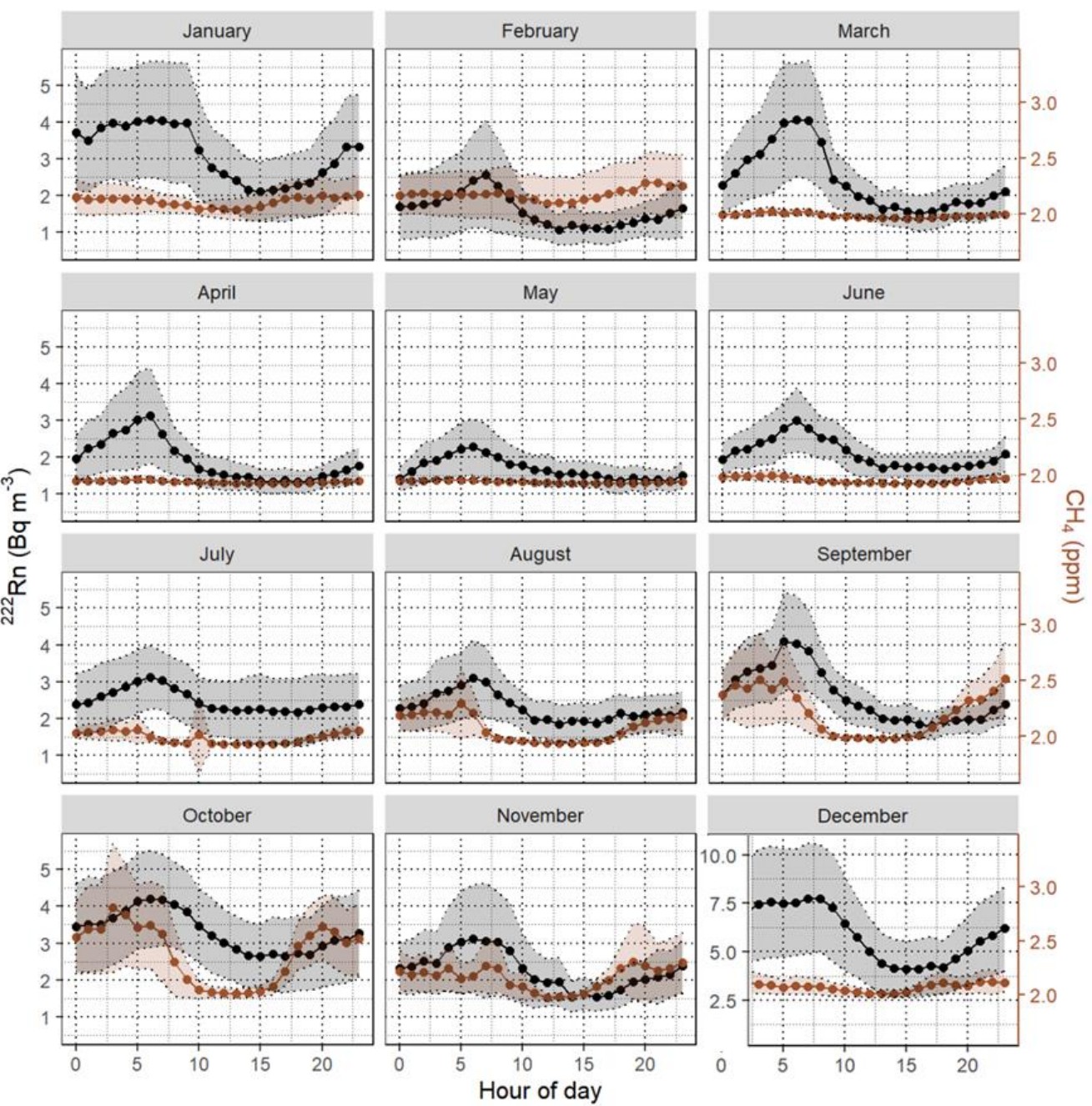

**Figure 5. Observed average monthly diurnal cycles for atmospheric radon (black line in Bq m$^{-3}$) and methane (orange in ppm) at DEC within the 2013-2019 period dataset. Points are hourly averaged values. The shaded area is the standard deviation for both CH$_4$ and $^{222}$Rn. Note the different $^{222}$Rn concentration scale for December.**

Figure 6 shows average monthly wind roses for the period 2013-2019 elaborated using wind speed and direction measured at DEC tower (10 m a.s.l.). From the multiple plots, two main seasonal patterns in wind regime are observed. During winter months (November to March) strong north-western winds coming from the Ebro valley are predominant and in summer the predominant winds are softer sea breezes coming from south, in agreement with Cerralbo et al. (2015). This last observation may indicate that during the summer months the local source term of the two gases may have a larger impact on the observed concentrations and thus on their diurnal cycles than during the winter months.

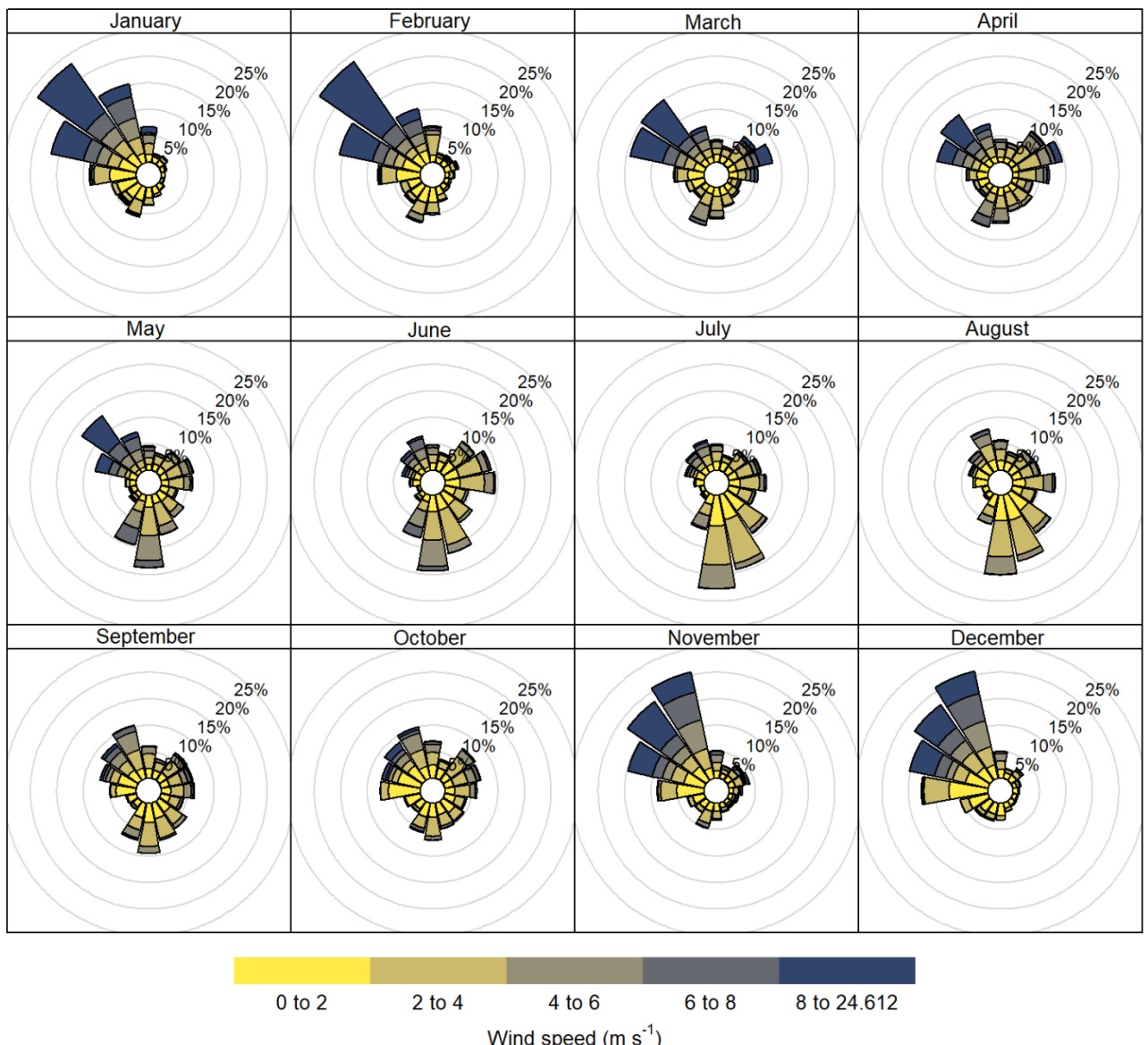

**Figure 6. Monthly wind roses for the upcoming winds at DEC station (10 m a.g.l.) within the 2013-2019 period dataset.**

Figure S4 of the Supplementary material presents monthly wind roses calculated for nighttime (21h UTC to 03h UTC, same windows as for RTM) and midday (11h UTC to 17h UTC). The southern winds are only present at midday and in warm months (May-October), as they are caused by the sea-land breeze. Northwestern winds in winter (November-March) are present day

and night, although they are stronger at nighttime. In spring (April-May) and fall (September-October), soft land-sea breezes

at night and sea-land breezes during the day can be observed, although the signal is weak.

## 3.2 Radon flux term evaluation

To assess the reliability of RTM-based $CH_4$ estimated fluxes, we have previously evaluated the radon flux term by assessing the performance of meteorological (WRF) and transport (FLEXPART-WRF) models. We examined also the footprint and the annual cycles of radon flux.

### 3.2.1 Meteorological model evaluation

Figure 7 shows a comparison between whole day and nocturnal (21h UTC to 03h UTC) wind patterns at DEC station for 2019 both from experimental observations and WRF surface field outputs. Although direction patterns are quite similar, modeled winds seem to be stronger. The model seems to overestimate the wind speed with an average bias of 2.0 m s$^{-1}$. The correlation factor found between simulated and observed wind speed is 0.57, and the circular correlation for wind direction is 0.52. The

model seems to better simulate temperature and pressure, as the correlation between these simulated variables and the observed values at DEC station are 0.89 and 0.92, respectively. It must be taken in consideration that the ERD is in a flat coastal zone with a quite complicated wind regime due to the Ebro valley channeling and the land-sea breezes, making the wind regime simulation a challenge for weather models, as reported in previous studies (Cerralbo et al., 2015).

However, looking only at the nocturnal window used for the RTM (21:00 – 03:00 UTC), the bias between the simulated and

the observed wind speed decreases to 1.7 m s$^{-1}$, but the correlation factor remains the same as for the whole day comparison. The RMSE for wind speed is 3.3 m s$^{-1}$ for the whole day and 3.2 m s$^{-1}$ for the nocturnal window. In Figure S5 of the supplement, the RMSE, bias and correlation between wind speed measurements and model across the different months is plotted, differentiating between whole day and nocturnal RTM window values. No significant differences were observed between nocturnal RTM window values and whole day intercomparison values. November is the month with higher correlation, but

also higher bias and RMSE, probably due to higher wind values, as observed in Figure 6.

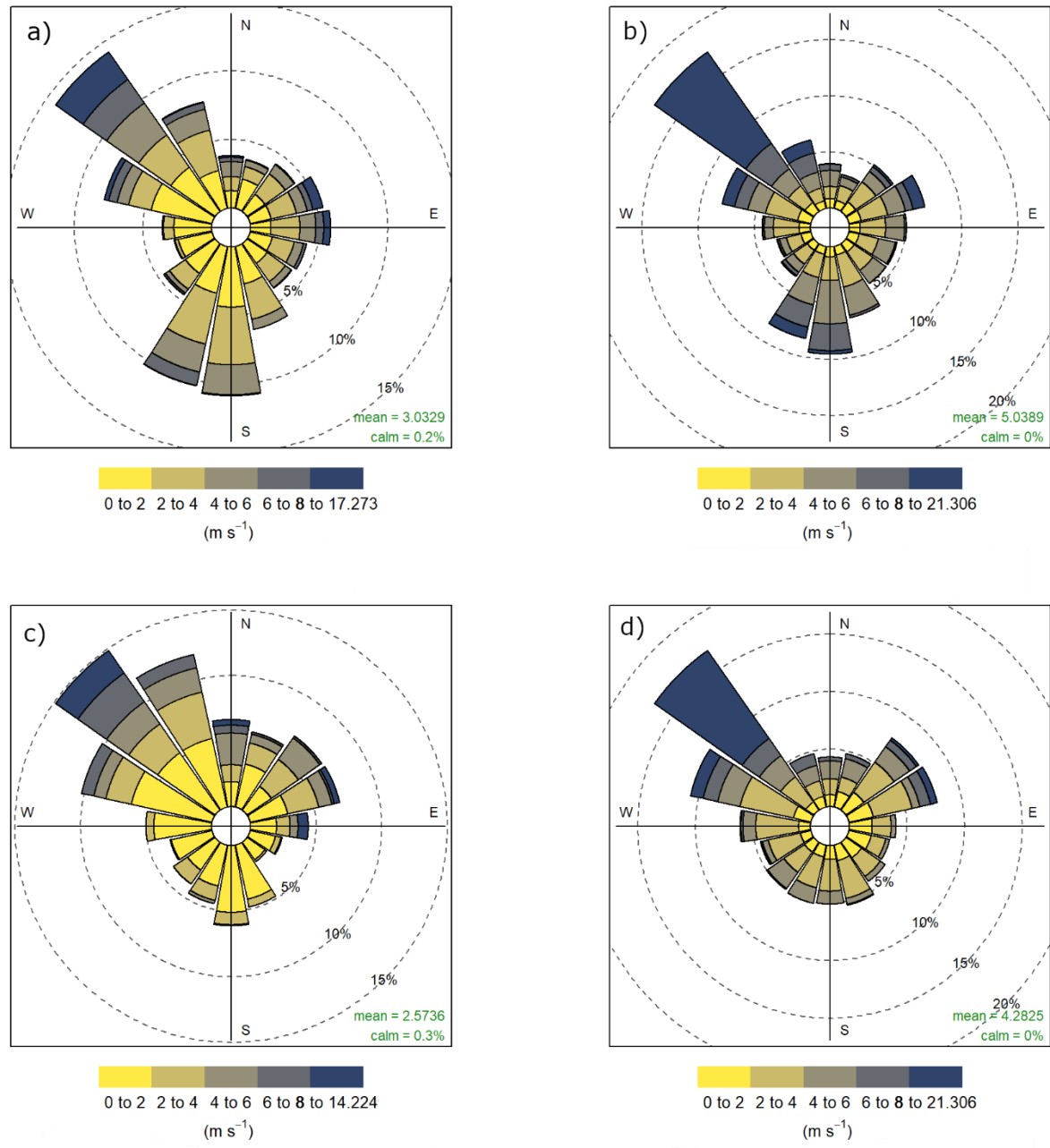

**Figure 7. Annual average wind roses at DEC within the 2019 dataset: a) observations over the whole day; b) modeled over the whole day; c) observations over the nocturnal window (21h UTC to 03h UTC); d) modeled over the nocturnal window (21h UTC to 03h UTC).**

### 3.2.2 Atmospheric transport model evaluation

The results of the quantitative evaluation of the performance of the models in simulating hourly atmospheric radon concentrations at DEC station during 2019 are shown in Table 1. It was examined by comparing simulated hourly radon concentrations, obtained with the same atmospheric transport model but different radon flux maps, against observed values at the same time. The smallest bias in the comparison of observed values against models was found within the Flex-WRF-GLDAS (-0.024 Bq m$^{-3}$) time series, and the best correlation with Flex-WRF-ERA5 (0.43). The WRMSE is similar for all three models. In the comparison model-to-model, the correlation coefficient between Flex-WRF-GLDAS and Flex-WRF-ERA5 outputs is 96%, the WRMSE is 0.21 Bq m$^{-3}$ and the bias is 0.29 Bq m$^{-3}$. In the case of Flex-WRF-Const and Flex-WRF-ERA5 the correlation coefficient is 85%, and the bias and the RSME are higher (-0.40 and 0.85, respectively) than those obtained comparing Flex-WRF-GLDAS and Flex-WRF-ERA5. The RSME in these two previous comparisons are much lower than those obtained when models' outputs are compared with measurements. This fact may indicate that the influence of the radon exhalation maps input is less significant than that of the atmospheric transport model or the meteorological model (as observed in section 3.2.1, for example, in the case of the wind).

**Table 1. Models' performance metrics based on the comparison of models' predictions against observed values and on the comparison between models.**

| Statistics | Flex-WRF-ERA5 vs. Observations | Flex-WRF-GLDAS vs. Observations | Flex-WRF-Const* vs. Observations | Flex-WRF-GLDAS vs. Flex-WRF-ERA5 | Flex-WRF-Const* vs. Flex-WRF-ERA5 |
|---|---|---|---|---|---|
| Bias (Bq m$^{-3}$) | -0.32 | -0.024 | -0.72 | 0.29 | -0.40 |
| R | 0.43 | 0.38 | 0.40 | **0.96** | **0.85** |
| RMSE (Bq m$^{-3}$) | 1.68 | 1.75 | 1.73 | 0.48 | 0.85 |
| WRMSE (Bq m$^{-3}$) | 0.61 | 0.62 | 0.64 | 0.21 | 0.34 |

*Constant value of 15.8 mBq m$^{-3}$ s$^{-1}$ on land pixels and 0 mBq m$^{-3}$ s$^{-1}$ on sea pixels.

When analyzing the statistical metrics shown in Table 1 for the different available time periods (see Table S1 of the Supplementary material), it was observed that in October-November the best fit between measured and simulated radon concentration values was with the ERA5 radon map, yielding an R value of 0.46 and a WRMSE of 0.61 Bq m$^{-3}$. When using a constant exhalation flux value, a lower R value was obtained (0.33) and a similar WRMSE value (0.63 Bq m$^{-3}$). In July-August period the fitting between models and observations was better, with correlation coefficients of 0.51, 0.53, and 0.58 for Flex-WRF-ERA5, Flex-WRF-GLDAS, and Flex-WRF-Const, respectively. Bias with Flex-WRF-ERA5 during July-August and October-November periods was -0.05 and -0.52 Bq m$^{-3}$, respectively, while bias with Flex-WRF-GLDAS was 0.22 and -0.23 Bq m$^{-3}$, respectively. However, bias with Flex-WRF-Const was quite high (-0.82 and -0.92 Bq m$^{-3}$ for both periods, respectively). Thus, although a similar RMSE and even a better correlation coefficient was obtained over this period using

Flex-WRF-Const, the constant radon exhalation map was finally excluded for the application of the RTM due to the higher bias observed during these two periods (those in which larger methane concentrations were measured).

Figure 8 shows a comparison between observed and simulated hourly radon atmospheric concentrations for the months of July and August 2019. Plots depicting the observed and modelled time series for other months in 2019 can be found in Figure S6 of the Supplementary material. In general, the model is able to reproduce the daily and synoptic radon variability over the different periods. However, differences can be observed during some synoptic episodes.

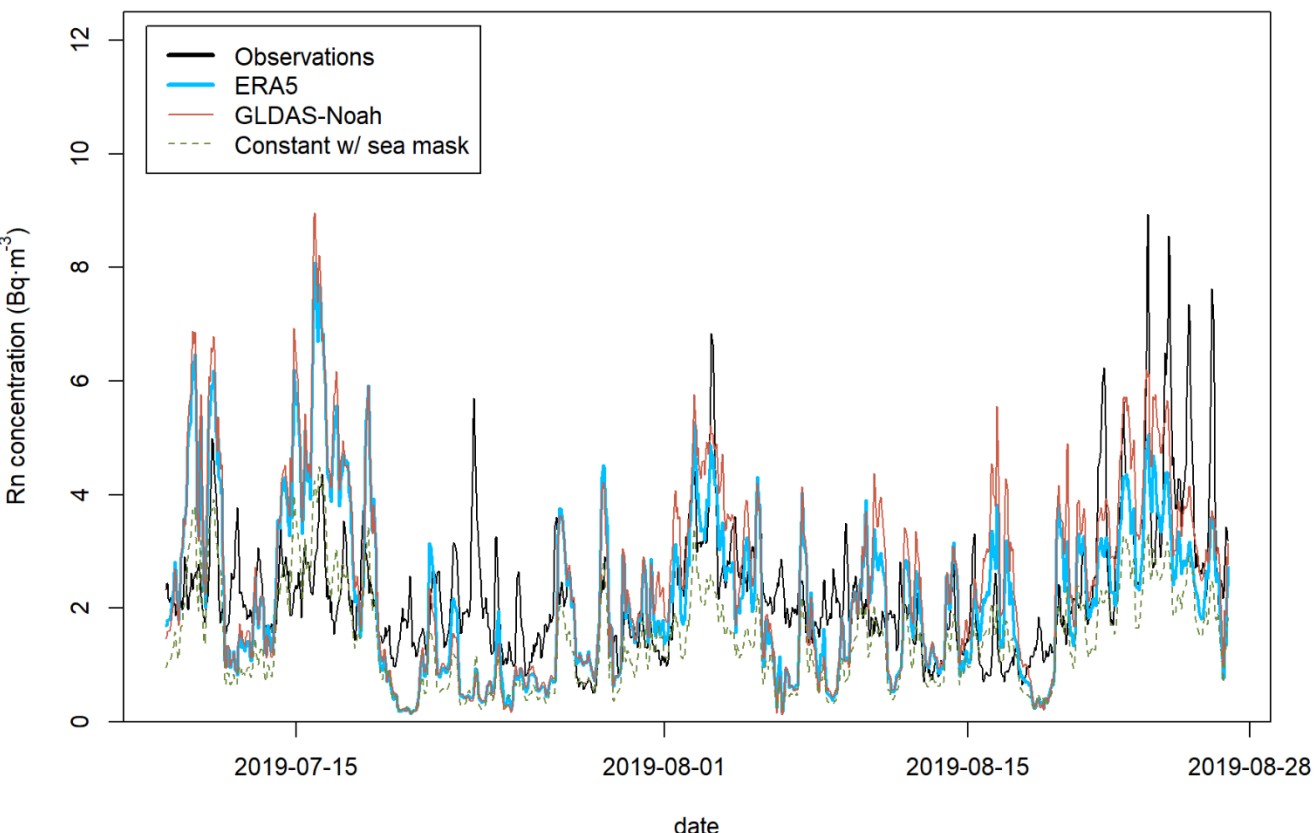

**Figure 8. DEC [222]Rn concentrations: observations (black line), modelled values using different radon exhalation maps: Flex-WRF-ERA5 (blue), Flex-WRF-GLDAS (red) and Flex-WRF-Const (green, dashed).**

A more detailed visual inspection of the simulated results seems to show that the model does not detect the remarkably high concentrations obtained during nocturnal peaks (accumulation phase), as seen in the periods of 29/03/2019 to 03/04/2019,

19/08/2019 to 27/08/2019 or 12/10/2019 to 29/10/2019 (Figure S6). Looking at the average diurnal cycle for the whole 2019 dataset (Figure 9) it can be noticed that observations and models peak at the same time (06h UTC), but the observed peak is much stronger than the simulated one, being 0.7 Bq m$^{-3}$ higher than Flex-WRF-GLDAS and 0.9 Bq m$^{-3}$ higher than Flex-

WRF-ERA5. However, between 10h UTC and 0h UTC the averaged radon concentrations are similar to the modelled ones, and the observed hourly average value remains between the Flex-WRF-GLDAS and the Flex-WRF-ERA5 hourly average values. The averaged modelled values using Flex-WRF-Const are much lower for all the diurnal cycle.

This bias between the observed and modelled radon concentrations at the daily peak was not constant over the tested year. For example, in April and May no bias was observed between radon observations and Flex-WRF-ERA5 modelled radon data. An

average bias of 1.21 Bq m$^{-3}$ was found in the months of October-November. This variability may also induce biases in the calculated nocturnal radon fluxes and therefore in the methane fluxes retrieved with the RTM. However, the variability in bias may not be solely attributed to the calculated radon fluxes but also to the WRF input, making it difficult to quantify. This could warrant further analysis.

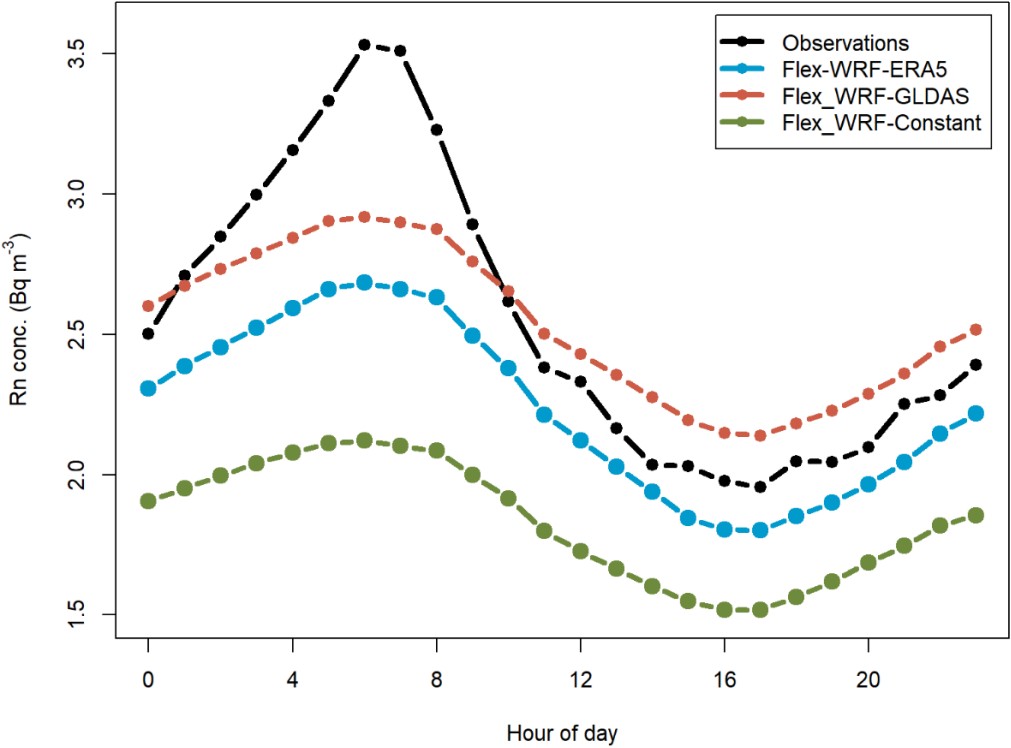

**Figure 9. Average diurnal cycle of $^{222}$Rn concentration at DEC: observations (black line), modelled values using different radon exhalation maps: Flex-WRF-ERA5 (blue), Flex-WRF-GLDAS (red) and Flex-WRF-Constant (green).**

It is known from the literature that the nocturnal boundary layer height (BLH) is one of the most challenging variables to simulate in mesoscale models (García-Díez et al., 2013) and this fact can cause a significant impact on transport models outputs (Díaz-Isaac et al., 2018; Gerbig et al., 2008; Mohan and Gupta, 2018). It has been proven in previous studies that the nocturnal

boundary layer often gets overestimated or underestimated in dispersion models (Arnold et al., 2010; Williams et al., 2011) and that it can be the main cause of divergence between simulated and observed nocturnal atmospheric concentrations. However, overall differences between model and observations may be attributable to both radon flux maps and transport models and from the data obtained it is not possible to attribute a higher contribution in the uncertainties to the radon maps or to the atmospheric models.


### 3.2.3 Nocturnal footprint of the station

At DEC station the inlet was located at 10 m a.g.l. and, therefore, it can be considered that the station footprint at night was very local. From the hourly footprints calculated with Flex-WRF for all nights where RTM was applied using the wind speed threshold of 1.5 m s$^{-1}$, the influence area was calculated too. Figure 10 shows the normalized average residence time for all

back trajectories from DEC during nights where the RTM was applied considering a 1.5 m h$^{-1}$ threshold. Results show that the ERD represents ~50% of the influence area for the air sampled at DEC station, while another 35% is over the sea, and the rest (15%) is a continental influence. Considering negligible radon and methane fluxes coming from the sea (Weber et al., 2019; Wilkening and Clements, 1975; Zahorowski et al., 2013), it may be considered that RTM-based $CH_4$ fluxes will mainly be due to the ERD contribution except for a 15% of continental influence. Taking into consideration that the models overestimate the

nocturnal mixing (as seen in Figure 9) and the nocturnal wind speed (as seen in Figure 7), the continental contribution would probably be lower. In addition, Figure 3 shows that methane emissions from this continental area, as presented by EDGAR inventory, are less than 0.5 mg $CH_4$ m$^{-2}$ h$^{-1}$. This represents only 10% of the methane flux values reported for the station's closest grids.

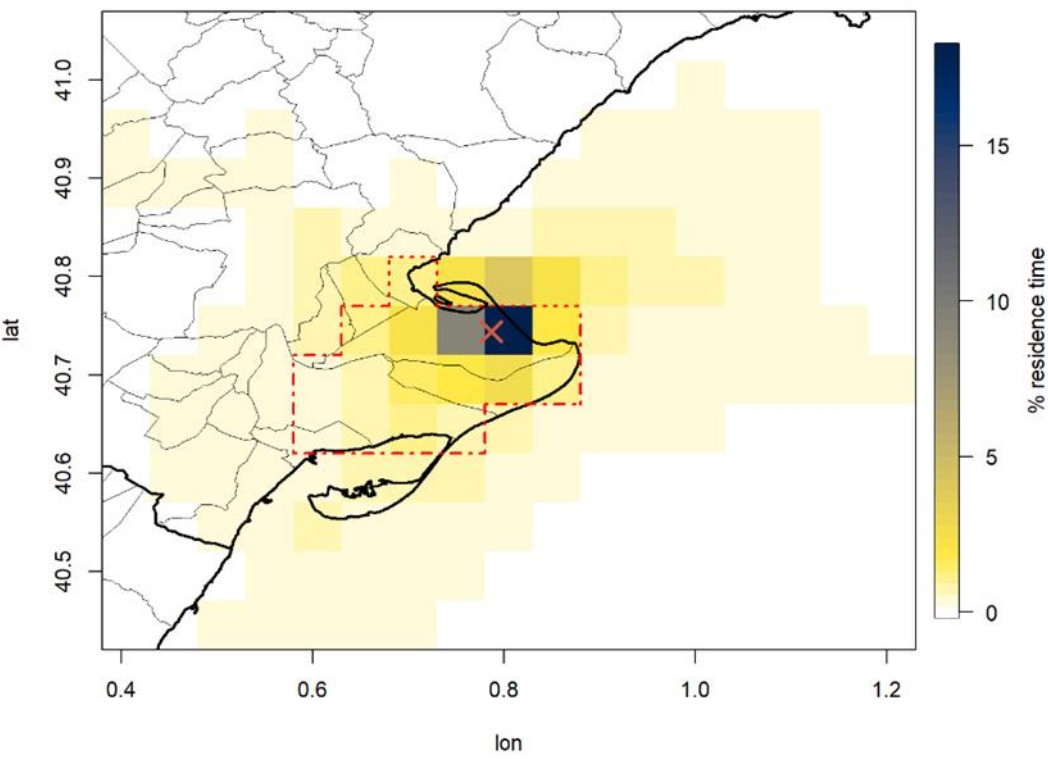

**Figure 10. Normalized average residence time of air masses arriving at DEC station (red cross) during RTM windows (21h UTC - 03h UTC) considering the with the 1.5 m s$^{-1}$ threshold. Pixels considering ERD are enclosed within the red dotted line.**

### 3.2.4 Radon flux cycles

Annual cycles of radon flux were calculated considering four different integration levels: (i) DEC station (the darkest pixel as illustrated in Figure 10); (ii) the whole ERD (red dotted line pixels as illustrated in Figure 10); (iii) the 70 km x 70 km window surrounding DEC (complete area shown in Figure 10), and (iv) the footprint-weighted radon flux applied for the RTM. As shown in Figure 11a, a clear annual cycle was observed with maximum values in summer. However, it seems that radon exhalation models are not taking into account an agricultural practice characteristic of rice paddies: the flooding of the fields (Figure 2) or, in other words, the existence of a water table of a certain level in rice crop fields that would decrease radon exhalation. Additionally, the increase in $^{222}$Rn concentration in winter months (in day and nighttime) (see Figure 5) does not seem compatible with the modeled radon flux cycle.

The observed bias between observations and modelled radon concentrations for different periods of the year (2019) are shown in Figure 11b, both for the whole day and only for the afternoon (15:00 to 18:00). Results show that both ERA5 and GLDAS radon exhalation maps are probably underestimating radon fluxes in fall (Oct.-Nov.), while they seem to overestimate radon fluxes in May. A constant continental value of 15.8 mBq m$^{-2}$ h$^{-1}$ seems to be underestimated at least for summer and fall

months, as was already observed analyzing the average diurnal cycle (Fig. 9). Therefore, the constant value $^{222}$Rn map will not be used to retrieve methane fluxes with RTM. Radon concentration bias in the afternoon does not differ from bias for the whole day.

Finally, the observed biases may indicate that the seasonality observed using radon exhalation maps may not agree with the real radon emission at ERD area. Although no bias data is available for December, high atmospheric radon concentration

values in that month (see section 3.1) indicate an increase in the radon flux for that month near DEC station, which is not observed based on radon exhalation maps. This increase could be driven by the complete drying of rice fields, which is not taken into consideration in the land models.

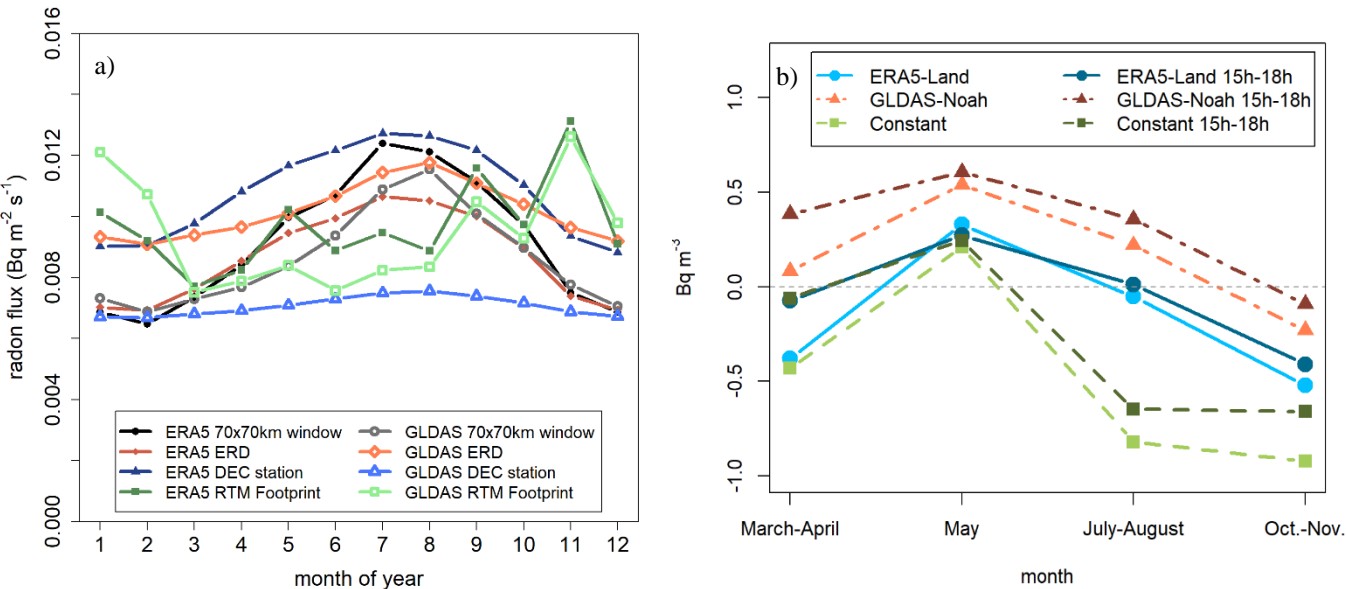

**Figure 11. a) Average annual cycle of $^{222}$Rn exhalation values for the 70 km x 70 km window, ERD and DEC station grid and RTM footprint-weighted for both ERA5 and GLDAS models; b) Bias between observations and modelled radon concentrations for 2019**
**for the whole day and for the afternoon (15h-18h).**

The footprint-weighted radon fluxes show a different trend than radon flux maps, which may be caused by the seasonality of winds, coming from the northwest in winter months. Overall, the RTM footprint-weighted radon fluxes are similar for both models and in the same order of magnitude as the 70 km x 70 km window or the ERD.

### 3.3 RTM-based CH$_4$ fluxes at DEC station.

Figure 12 presents the monthly median values of the nocturnal CH$_4$ fluxes obtained using the RTM with two radon exhalation maps: ERA5 and GLDAS taking into consideration the whole dataset (left plot) and only nights where the average wind speed was below 1.5 m s$^{-1}$ (right plot). Constant radon flux map was not used in this second part of the study as previously explained (see section 3.2.2).

The Shapiro-Wilk test for normality (Shapiro and Wilk, 1965) was performed to the RTM CH$_4$ flux data indicating that flux
data was not following a normal distribution (w <0.29, p-value < 0.01), but that there were no reasons to reject a log-normal distribution (w = 0.99, p-value = 0.67). Therefore, monthly median CH$_4$ flux values were used instead of monthly mean values. The RTM selection criteria used in this study restricted the percentage of nocturnal accumulation events used over the 7 years (2013-2019) dataset to its 15% when using the whole dataset and to 7% when using only nights with average wind speed below 1.5 m s$^{-1}$. Table 2 shows, for each month, the percentage of nights that were selected over the total available, the median and
the standard deviation of the log-normal distribution ($\hat{\sigma}$) of the retrieved RTM-based CH$_4$ fluxes using the two radon exhalation maps. Table 3 shows the same values when using only events with an average wind speed below 1.5 m s$^{-1}$.

**Table 2. Statistics of the RTM application at DEC station for the 2013-2019 period.**

| Month | N° of nights with measurements | N° of nights eligible for RTM | Percentage (%) | Median CH$_4$ flux (ERA5) (mg CH$_4$ m$^{-2}$ h$^{-1}$) | $\hat{\sigma}$ (ERA5) | Median CH$_4$ flux (GLDAS) (mg CH$_4$ m$^{-2}$ h$^{-1}$) | $\hat{\sigma}$ (GLDAS) |
|---|---|---|---|---|---|---|---|
| January | 42 | 8 | 19% | 2.00 | 1.21 | 2.50 | 1.26 |
| February | 28 | 2 | 7% | 1.92 | 1.13 | 2.14 | 1.09 |
| March | 57 | 12 | 21% | 0.38 | 1.14 | 0.38 | 1.13 |
| April | 68 | 17 | 25% | 0.21 | 1.09 | 0.19 | 1.19 |
| May | 54 | 13 | 24% | 0.64 | 0.69 | 0.51 | 0.73 |
| June | 49 | 8 | 16% | 1.78 | 0.96 | 1.38 | 0.97 |
| July | 86 | 15 | 17% | 2.70 | 1.05 | 2.20 | 1.04 |
| August | 71 | 14 | 20% | **4.76** | 0.88 | **4.45** | 0.93 |
| September | 44 | 9 | 20% | 2.95 | 1.66 | 2.60 | 1.75 |
| October | 74 | 10 | 13% | **13.90** | 2.44 | **14.38** | 2.43 |
| November | 38 | 5 | 13% | **4.08** | 0.58 | **5.51** | 0.56 |
| December | 68 | 16 | 24% | 0.34 | 0.91 | 1.02 | 0.85 |
| **Total** | **679** | **98** | **15%** | **3.0** | **1.59** | **3.1** | **1.62** |

**Table 3. Statistics of the RTM application at DEC station for the 2013-2019 period considering only nights with average wind speed < 1.5 m s$^{-1}$.**

| Month | N° of nights with measurements | N° of nights eligible for RTM | Percentage (%) | Median CH$_4$ flux (ERA5) (mg CH$_4$ m$^{-2}$ h$^{-1}$) | $\hat{\sigma}$ (ERA5) | Median CH$_4$ flux (GLDAS) (mg CH$_4$ m$^{-2}$ h$^{-1}$) | $\hat{\sigma}$ (GLDAS) |
|---|---|---|---|---|---|---|---|
| January | 42 | 2 | 5 % | 3.00 | 1.98 | 3.82 | 2.15 |
| February | 28 | 0 | 0 | - | - | - | - |
| March | 57 | 4 | 7 % | 0.66 | 0.99 | 0.55 | 0.87 |
| April | 68 | 2 | 3 % | 0.37 | 0.77 | 0.4 | 0.83 |
| May | 54 | 1 | 2 % | 0.31 | - | 0.24 | - |
| June | 49 | 3 | 6 % | 1.40 | 0.23 | 1.09 | 0.22 |
| July | 86 | 7 | 8 % | 3.11 | 1.18 | 2.54 | 1.13 |
| August | 71 | 8 | 11 % | **4.33** | 0.31 | **3.81** | 0.4 |
| September | 44 | 5 | 11 % | 6.79 | 0.98 | 6.25 | 1.04 |
| October | 74 | 5 | 7 % | **13.90** | 0.93 | **14.38** | 0.87 |
| November | 38 | 1 | 3 % | **7.07** | - | **6.30** | - |
| December | 68 | 7 | 10 % | 0.69 | 0.79 | 0.74 | 0.71 |
| **Total** | **679** | **45** | **7 %** | **3.5** | **1.36** | **3.4** | **1.34** |

Although some differences are observed when using only night events when the wind is lower than 1.5 m s$^{-1}$ compared to when no wind restriction is used, the methane fluxes follow the same seasonal pattern. In October, the month with the largest estimated emissions, flux values are exactly the same. The largest discrepancies are found in September, when methane fluxes with wind restrictions more than doubles the methane fluxes measured with no wind restrictions. The average annual fluxes are 15% higher when only the events with a wind speed under 1.5 m s$^{-1}$ are selected.

The number of eligible events with wind restrictions is less than half of the nights when no wind restrictions are used. Therefore, the representativeness decreases and in February, for example, no events are available as eligible for RTM. Thus, and given the small differences using or not using the wind speed threshold value, the values obtained considering no wind restriction will be used for the comparison of the annual emissions with other studies and inventories.

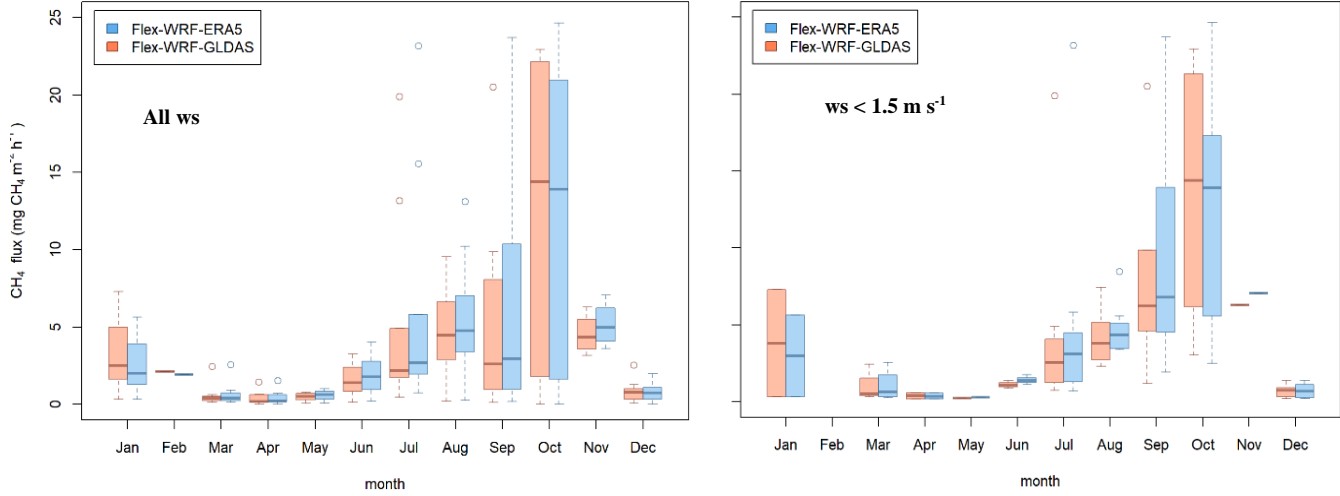


**Figure 12. Boxplots of the methane RTM based fluxes over the DEC station calculated using Flex-WRF-ERA5 (blue) and Flex-WRF-GLDAS (red) for every month of the year (period 2013-2019) for the whole dataset (left) and only taking into consideration events with an average wind speed (ws) < 1.5 m s⁻¹ (right). Outliers are represented with circles, boxes represent the region between interquartiles Q1 and Q3, and horizontal solid lines the medians.**

The average difference between RTM-based CH$_4$ fluxes obtained using Flex-WRF-ERA5 and Flex-WRF-GLDAS models was only $0.1 \pm 0.2$ mg CH$_4$ m$^{-2}$ h$^{-1}$. This difference remains the same regardless of whether the wind threshold is applied. The monthly methane RTM based fluxes obtained with Flex-WRF-ERA5 (and Flex-WRF-GLDAS) showed a strong seasonality, with a maximum in October with a median flux of 13.9 (14.4) mg CH$_4$ m$^{-2}$ h$^{-1}$ and a minimum between the months of March, April and May, with values between 0.2 and 0.7 (0.2 and 0.6) mg CH$_4$ m$^{-2}$ h$^{-1}$. For the data without wind threshold, the average

methane flux for the period between tillering and straw incorporation (i.e., between June and November) was 5.2 (4.9) mg CH$_4$ m$^{-2}$ h$^{-1}$. For the period when the fields were dried (i.e., February to April), the average methane flux decreased to 0.8 (0.9) mg CH$_4$ m$^{-2}$ h$^{-1}$. Finally, the total annual average emission rate was calculated to be 3.1 (3.0) mg CH$_4$ m$^{-2}$ h$^{-1}$. This emission is equivalent to an annual emission of 262.8 kg CH$_4$ ha$^{-1}$.

Based on radon concentration simulation results, it was not possible to determine which of the two radon flux maps performed

better, as results varied for the different periods of the year and showed a consistent trend between both. In addition, no previous studies are available in the literature evaluating these two radon flux maps. Therefore, for the calculation of methane fluxes with RTM, both maps have to be considered. However, differences in flux measurements are low (< 5%) and, therefore, we will refer only to the results obtained with ERA5 radon exhalation map for the comparison with other studies.

### 3.4 RTM-based CH$_4$ fluxes *vs*. CH$_4$ fluxes from the literature

Average 2013-2019 methane fluxes from EDGAR v7.0 inventory are shown in Figure 3. Although EDGAR v7.0 counts for agriculture soils emissions, its data does not consider the emissions from rice paddy fields at ERD, as the emissions from agricultural soils assigned in the pixels of the ERD are below 0.02 mg CH$_4$ m$^{-2}$ h$^{-1}$. The 80% of the emission assigned at the pixel of DEC station is related to a cow farm located 9 km west from the station. The largest emissions in the area are located 55 km Northeastern of the sampling site, and are related to a petrochemical industry complex. From this dataset, it may be

confirmed that no accounted significant anthropogenic methane emissions are present in the area around the station apart from the unaccounted methane due to ERD rice fields. The fact that from January to June no methane diurnal cycles are observed at DEC supports the hypothesis that the methane fluxes observed can be entirely attributed to the local rice fields. Moreover, assuming zero emissions of methane and radon from the sea, it can be inferred that when wind is coming from the sea all the signal in RTM is coming from the ERD. Therefore, taking into consideration the footprint area, RTM results can be considered

as a good proxy of the variability of methane emissions due to rice cultivation cycle in ERD over the months of the year.

    Monthly methane flux values obtained by Martínez-Eixarch et al. (2018) at ERD with static chambers are plotted together with RTM based results from the present work in Figure 13. The plot shows a remarkable correlation between both results, obtained using independent methodologies. The seasonal variability of the flux estimated using both methodologies follows a consistent pattern during productive and fallout months. However, December stands out as the month with the greatest disagreement

between the two methodologies, with the RTM estimation being 2 mg CH$_4$ m$^{-2}$ h$^{-1}$ lower than the static chambers estimation. This disparity could be caused by an underestimation of the radon fluxes in December, as seen in section 3.2.4. The absolute values are similar during months with the highest emissions. For instance, in October RTM-based results estimated a median flux of 13.9 mg CH$_4$ m$^{-2}$ h$^{-1}$, while the fluxes from static chambers were calculated as 14.7 ± 4.2 mg CH$_4$ m$^{-2}$ h$^{-1}$.

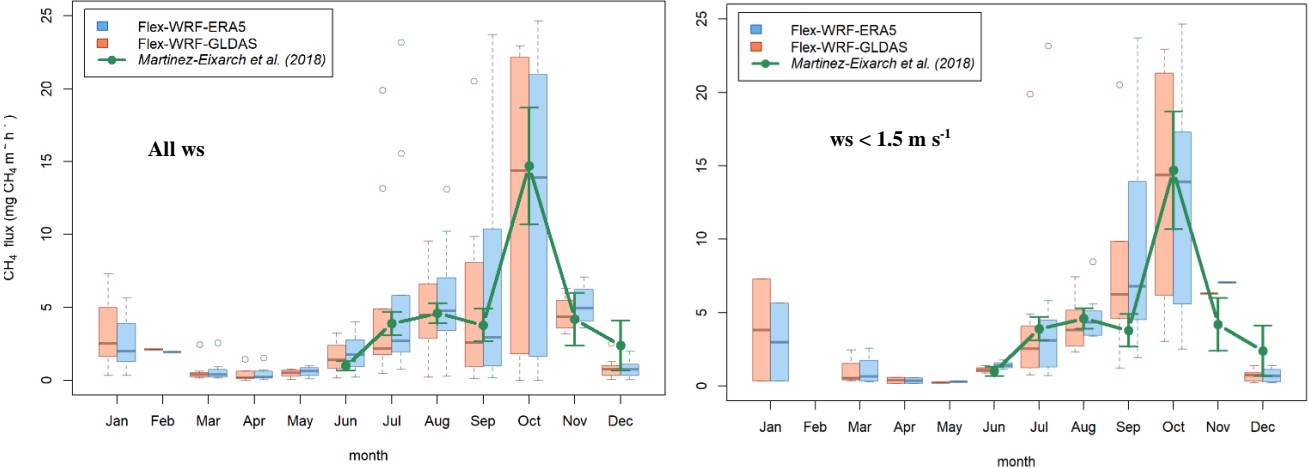

Figure 13. Boxplots of the methane RTM based fluxes over the DEC station calculated using Flex-WRF-ERA5 (blue) and Flex-WRF-GLDAS (red) for every month of the year (period 2013-2019) for the whole dataset (left) and only takin in consideration events with wind seed < 1.5 m s⁻¹ (right), and methane fluxes over ERD using static chambers (green points and lines) (Martínez-Eixarch et al., 2018).

In a study conducted by Wang et al. (2018), a global modelling of rice fields emissions was undertaken, accounting for multiple parameters at each country. The country-specific emission factor for Spain was estimated to be 1.13 kg $CH_4$ h⁻¹ day⁻¹. In the Ebro Delta, the emission factor according to the latest national inventory of Spain reported to the UNFCC (National Inventory Report of Spain, 2023), is 1.32 kg $CH_4$ h⁻¹ day⁻¹. This inventory emission corresponds to an emission of 5.54 mg $CH_4$ m⁻² h⁻¹ during the crop period and a total yearly emission of 198.5 kg $CH_4$ ha⁻¹ using the inventoried rice crop period of 150 days reported to the UNFCC. With the RTM methodology the estimated annual emission was 262.8 kg $CH_4$ ha⁻¹. In the work by Martínez-Eixarch et al. (2018), the annual emission is reported to be 262.6 kg $CH_4$ ha⁻¹. The outstanding similarity using both methodologies, however, has to be taken carefully due to the high uncertainty of both methods, but it confirms the suitability of both methodologies for the calculation of methane fluxes in a region as the ERD.

RTM-based $CH_4$ flux estimations show that emissions were distributed along the year and that the higher ones corresponded to the months of harvest and straw management. However, during all months that the fields were flooded, emissions were significantly higher than those inventoried in EDGAR, and only similar to the inventories in months where the fields are dried (March-April). As in the work by Martínez-Eixarch et al. (2018), it can be seen that neglecting the fallow season can significantly underestimate annual emissions. Methane emissions from October to December account for 54% of the total, while emissions during the growing period (May-September) represent only 31%. Finally, emissions from January to April make up just 15% of the total methane emissions.

### 3.5 Uncertainty and representativeness of the RTM-based CH₄ fluxes at DEC

One of the simplifications considered for the application of the RTM is that fluxes should be homogeneous around the station, as strong point sources could affect the concentrations measured at the station and the regional flux estimation. Although the ERD is relatively small compared to other areas where the RTM has been applied (Schmidt et al. 1996, Levin et al. 2021), the short height of the sampling tower and the homogeneity of the ERD makes this area well-suited for the application of the RTM.

The difficulty of the application of the RTM in this study is sure but the evaluation of both meteorological and atmospheric transport models for the area and periods of interest, the comparison of the RTM based methane fluxes results with national inventories and other independent experimental studies from the bibliography add useful highlights for the research in this field and improvement of the RTM application protocols.

As seen in section 3.2, the WRF model in regions like ERD does not always simulate correctly the nocturnal accumulation and the wind speeds, deriving to important bias in concentration simulations. The advantage of the work exposed here is that due to the short height of the sampling point, the footprint of the station is quite small, within a few kilometers, and thus, the footprint is more reliable. Moreover, by applying the 1.5 m/s wind speed threshold for selecting RTM-feasible night events, we ensure minimal advection from continental sources. In agreement with EDGAR emission inventories, Corine Land Cover map, and other bibliographic studies, there are no significant contributions to the methane emissions compared to those from rice paddies. Methane fluxes over the footprint area and when rice paddies are dried are between 0.3 and 0.6 mg CH4 $m^{-2}$ $h^{-1}$, while our results, corroborated by literature (Martínez-Eixarch et al., 2018) and other studies in rice fields (Wang et al., 2018), show methane fluxes due to the rice fields above 3 mg CH$_4$ $m^{-2}$ $h^{-1}$ between July and November. Considering these differences, the external contribution to the rice paddies methane emissions may be assumed with an uncertainty between 10-15%.

One of the elements that produces more uncertainty in the calculus of the methane flux using the RTM is the uncertainty of real radon flux distribution over the footprint area of the measurement station (Vogel et al., 2012, Levin et al., 2021). In Eq. (3), the radon flux is directly proportional to the estimated methane flux. Therefore, an error in the radon flux will proportionally produce an error in the estimated methane flux. For this reason, until the radon flux maps in the area over the DEC station could be sufficiently validated, the global annual methane flux estimated with the RTM should be carefully accepted assuming a certain uncertainty. On the other hand, the integration, in this work, of data from several years, makes the result more robust than if using data from a single year.

One of the limitations of the RTM is that only the nocturnal emissions are monitored. In the case of rice fields, it is well known that the gross ecosystem photosynthesis (GEP) and the soil temperature are drivers of CH$_4$ flux variability (Hatala et al., 2012). Although diel fluxes and nocturnal fluxes keep a strong correlation (Wassmann et al., 2018), methane emissions in the early

afternoon can be between 10% and 20% higher than the nocturnal emissions during the productive months (Alberto et al., 2014; Dai et al., 2019; Minamikawa et al., 2012). This difference may lead to an underestimation, ranging between 10% (Weller et al., 2015) and 20 % (Wassmann et al., 2018), of diel fluxes if considering only the nocturnal emissions.

## 4 Conclusions

Using currently available radon flux maps, the meteorological model WRF, modeled particle back trajectories with
atmospheric transport model FLEXPART, and methane and radon atmospheric concentrations observations from a 10 m a.g.l. tower, the methane fluxes variability over the rice crop area of the ERD, in the north-east of the Iberian Peninsula, was evaluated. Prior to this calculation, modeled back trajectories and the different radon exhalation maps used in the study were evaluated by simulating radon concentrations at the tower sampling point and comparing them with observations. The two main conclusions drawn from this previous comparison are:

1) Atmospheric transport models are not yet able to accurately estimate the nocturnal boundary layer in coastal areas such ERD, often overestimating the vertical mixing.

2) The seasonality observed in the radon exhalation maps from Karstens and Levin (2023) may not be adequately parametrized in the ERD area, as different bias among the months are observed between the modeled and observed atmospheric radon concentration, mainly during peak events. Although the biases could also be produced by the seasonal difference in the
transport model performance, the estimated radon fluxes at DEC do not seem to take into consideration the seasonality of the water table height within this area.

From the application of the RTM, a strong annual cycle of methane emissions is observed. This annual cycle is related with the rice crop cycle, with the highest emissions in October coinciding with harvest and straw incorporation into the fields. When a wind threshold of 1.5 m s$^{-1}$ is used for RTM application, minor differences are observed. The overall pattern is the same with
an increase of 15% in average annual emissions. The methane emission pattern and values are remarkably similar to a study done with static chambers along two years, by other authors, which can be useful to validate the presented methodology. The total annual methane emissions estimated are 262.8 kg $CH_4$ ha$^{-1}$, close to the 262.6 kg $CH_4$ ha$^{-1}$ from the study of the static chambers and 32% higher than the UNFCC inventoried value (198.5 kg $CH_4$ ha$^{-1}$). The independent EDGAR emissions database does not account for methane emissions from this rice fields area.

Absolute emission values given by the RTM may be handled with care, as there are many assumptions and simplifications considered. However, its application has been proven to be incredibly useful to know the inter-annual variability of regional methane emissions (Levin et al., 2021), to amend inventory values for not considering seasonality of livestock management

(Grossi et al., 2018), or, as in this work, to understand and quantify the seasonal variability of emissions over a reduced area and its relation with the different phases of agricultural processes such as the rice cultivation.

Due to the hostile environmental conditions at DEC station (extremely high humidity, elevated temperatures and salty air) the dataset presents several gaps. Thus, a year-to-year variability study was not feasible. Longer datasets in the future may be used with the RTM to monitor the inter-annual variability in methane emissions, which could be influenced by changes in agricultural management (e.g., straw management, water management, or fertilization changes), without requiring resource-intensive extended static chamber campaigns. Considering the resources needed, an atmospheric station equipped with radon

and methane instrumentation could be significantly more efficient than performing periodic surveys using accumulation chambers across the entire area of interest over an extended period.

**Appendix. Flex-WRF modeling parameters.**

Table A1 shows the main parameters used in the WRF modeling for radon simulation during 2019, as well as for the RTM footprint spanning the period from 2013 to 2019.

**Table A1. WRF parameters for Flex-WRF simulations.**

| WRF version | 4.2.1 |
|---|---|
| PBL Scheme | Yonsei University scheme |
| Microphysics | WRF Single-moment 6-class scheme |
| Surface Layer physics | Revised MM5 surface layer scheme |
| Horizontal resolution | d01: 27 km x 27 km |
| | d02: 9 km x 9 km |
| | d03: 3 km x 3 km |
| Vertical layers | 57 |
| Top of the atmosphere | 50 hPa |
| Meteorological initial conditions | ERA5 (Hersbach et al., 2020) |

Flex-WRF was parametrized to be used with mean winds from WRF output and with convection, turbulence and PBL schemes.

As for the domains, while in WRF a lambert conformal conic projection was used for a better performance, a regular lat-lon grid was used in Flex-WRF for an easiest merge with radon maps, which are in lat-lon regular grid projection.

Figure A1 shows the limits of the three domains for WRF simulations and the two domains for Flex-WRF back trajectories.

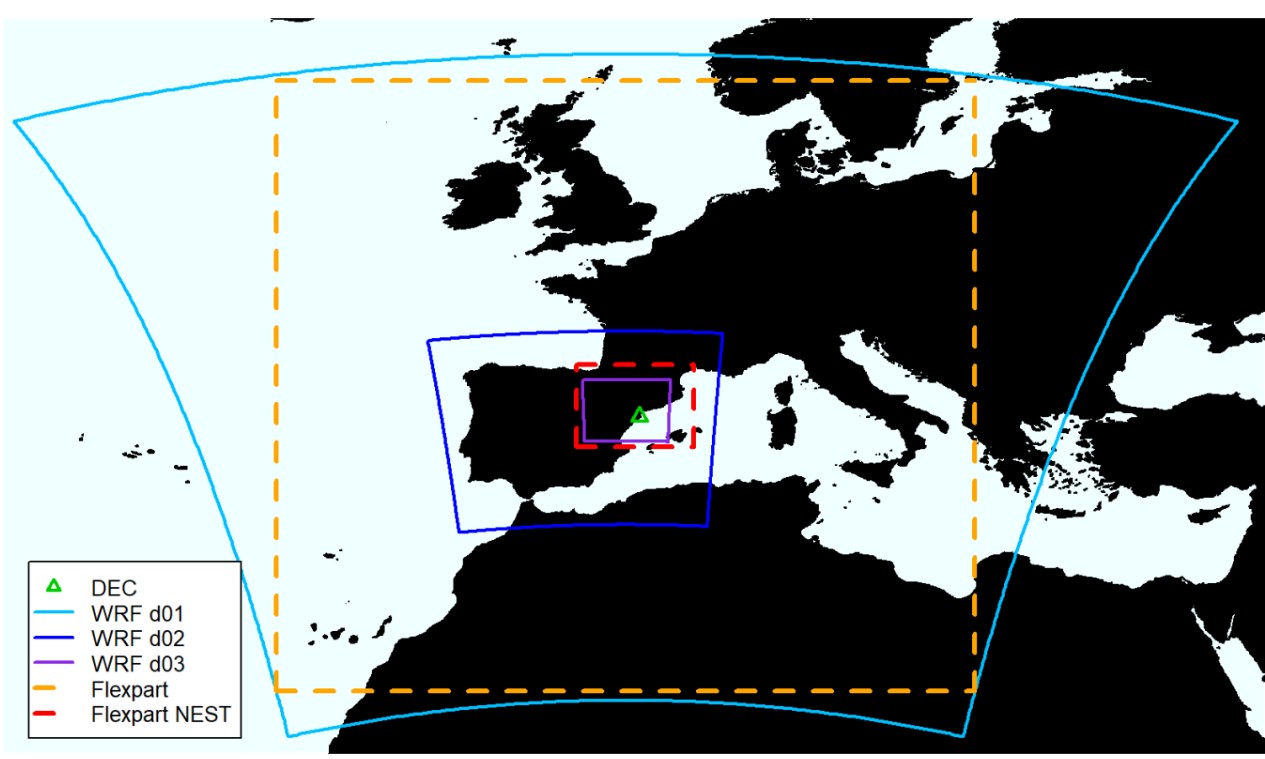


**Figure A1. WRF and Flexpart domains for Flex-WRF simulations.**

**Data availability**

The data and codes for this paper are available at the CORA Repositori de Dades de Recerca with DOI https://doi.org/10.34810/data1332.

**Author contributions**

J-AM, as PI of the ClimaDat project, designed and leaded the creation of the ClimaDat network. AÀ, SB, CG, LC, J-AM and RC participated actively in the mounting and maintenance of the DEC station atmospheric measurements and drying system. RC and CG were in charge of the Picarro and AMON instruments and data production, respectively. CG designed the RTM application study for DEC station. RC led this work, performed the Flex-WRF simulations needed for the RTM applications

as well as for the models evaluation and wrote the original draft of the manuscript. AV and CG coordinated the data analysis and discussions. AÀ, CG and AV contributed actively to the manuscript writing. All authors participated in the manuscript writing and agreed to the published version of the paper.

## Competing interests

The authors declare that they have no conflict of interests.

## Acknowledgements

A. Àgueda is a Serra Hunter Fellow. The authors would like to thank the members of the ClimaDat project who worked for the maintenance of the ClimaDat DEC station: Manel Nofuentes, Eusebi Vàzquez, Òscar Batet and Paola Occhipinti. The authors would also like to acknowledge the workers of the "Estació biològica del Canal Vell", particularly Miquel Àngel and Laura, for their help and cooperation. Finally, this work is dedicated to the beloved memory of our ClimaDat colleague Òscar
Batet Torrell.

## Financial support

All the measurements at DEC station were supported by the ClimaDat project [Obra Social "La Caixa", 2010 - 2019]. Part of the research was carried under the project "Methane interchange between soil and air over the Iberian Peninsula" (CGL2013-46186-R) funded by the Ministerio Español de Economía y Competitividad. The modeling, data analysis, application of the
RTM and manuscript writing was carried on under the 19ENV01 traceRadon project, which has received funding from the EMPIR program co-financed by the Participating States and from the European Union's Horizon 2020 research and innovation program.

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
