# Peer review of "Estimation of seasonal methane fluxes over a Mediterranean rice paddy area using the Radon Tracer Method (RTM)"

_EGUsphere, 2024_

## Referee Comment (RC3)

Review of Curcoll et al. (2024), **„Estimation of seasonal methane fluxes over a Mediterranean rice paddy area using the Radon Tracer Method (RTM)"**

This paper aims to estimate methane flux and its seasonal variability using the "local scale" Radon Tracer Method (RTM) in a rice paddy in the Ebro Delta. The manuscript is well-written and logically structured, with a clear presentation of methodology and results. Given the importance of methane emissions from rice paddies as a significant contributor to agricultural greenhouse gases, as highlighted in the introduction, this study is highly relevant for meeting the goals of the Paris Agreement.

However, before recommending publication, I would suggest further clarification on some major points.

**Methodology:**

The "near flat variability of radon" raises significant concerns in this study. The authors are conducting measurements at a site that remains flooded for much of the year, which would lead to no radon flux. In contrast, methane ($CH_4$) emissions may still occur from this area. Given that this is a coastal site, radon flux from the ocean is minimal, but coastal regions can still generate $CH_4$.

The local accumulation version of the Radon Tracer Method (RTM) assumes that the fluxes of both radon and methane are similarly distributed in space and homogeneously spread across the measurement footprint. This homogeneity is crucial for neglecting advective effects. Additionally, the "accumulation" model is valid only under very stable conditions, specifically when wind speeds near the surface are less than or equal to 1.5 m/s. If the wind speeds exceed this threshold, the correlations observed between radon and $CH_4$ would likely result from fetch effects rather than local accumulation *(This limitation is also addressed in greater detail by another reviewer, Scott Chambers.).*

To achieve their objectives more effectively, the authors would need to focus exclusively on low wind speed nights, which, based on their wind rose data, are quite rare. This restriction would significantly reduce their dataset. Furthermore, they could only apply their methodology during times when the rice paddies were not flooded, ensuring that radon flux was indeed present from the relevant fetch region.

**More detailed points in methodology:**

- The dataset spans from 2013 to 2019, providing approximately seven years of data with a total of 61,320 hourly measurements. *I would appreciate some clarification on the rationale for specifically including data from 2019.*
- However, only about 30% of this data (approximately 18,396 observations) was usable due to instrument maintenance issues. Additionally, limiting the analysis to just six hours per night further restricts the dataset, raising concerns about the representativeness of the flux measurements. For example, Table 2 shows that data from February across seven years includes usable measurements from only two nights.
- The diurnal variability of radon appears very flat, which may be attributed to the site's proximity to the coast and the limited radon fluxes present.
- Finally, the data selection criterion employed a threshold of $R^2 \geq 0.5$, which is relatively low; a minimum of 0.7 is generally preferred. This suggests that the observed diurnal radon signal may arise from air advected to the study site or from minimal contributions from exposed ground. During other times of the year, wind speeds are often quite high. When comparing

this distance to the scale of the rice paddy fetch in various directions from the measurement site, the implications for flux representativeness become even more pronounced.

I understand that you are using the Weather Research and Forecasting (WRF) model, which operates on a mesoscale, to estimate the influenced footprints (for a local). However, in section 3.2.1, regarding the evaluation of the meteorological model, it's noted that the correlation between simulated wind speed and observed wind speed is 0.57. Additionally, it appears that the model tends to overestimate wind speeds for most of the assessment period.

Given these findings, it is important to critically evaluate the accuracy of the model in estimating the footprint dimensions. Overestimating wind speed can lead to significant discrepancies in how gases are predicted to disperse. If the model is consistently overpredicting wind speeds, the resultant footprints may be smaller or inaccurately positioned, which could misrepresent the true spatial extent of influence.

Furthermore, it is essential to clarify whether this correlation of 0.57 applies uniformly across the entire study period from 2013 to 2019, or if it is specific to certain times of day, such as nighttime or daytime. Variations in wind patterns between day and night can significantly affect dispersion characteristics. If the correlation is weaker during certain periods, this could further impact the model's reliability in estimating footprint dimensions.

To improve the robustness of your analysis, consider the following:

1. **Temporal Analysis**: Investigate whether the correlation varies by time of day. This can provide insights into how well the model performs under different meteorological conditions.
2. **Sensitivity to Overestimation**: Assess how the overestimated wind speeds affect gases and footprint dimensions. This sensitivity analysis can help quantify the potential impacts of the model's inaccuracies.

**More specific comments:**

**L328:** I remain unconvinced that the average radon levels observed in December can be attributed solely to local radon fluxes. The high standard deviation associated with this mean suggests significant variability in the data, indicating that the observed radon levels may be more representative of a unique, one-time event rather than a consistent, ongoing trend driven by local sources. Could you please look in more detail for this?

**L422-427: "In the present work, the correlation between observed wind and modelled wind (0.52) is higher than the correlation between observed and modelled radon concentrations (0.38 – 0.43). Moreover, differences between the three radon exhalation models are much lower than between observations and models. Therefore, although no observational data was available on BLH for DEC station, it can be deduced that most of the disagreement between models and observations may have come from the nocturnal boundary layer simulation rather than by radon exhalation maps uncertainties."** – Could you clarify how you reached this conclusion? I'd like to understand the reasoning behind attributing most discrepancies to boundary layer simulations instead of the radon exhalation models. What specific evidence or analysis supports this interpretation? I'm not entirely convinced that relying solely on correlation coefficients provides a complete picture.

**L505-512:** The authors compare the RTM results from 2013 to 2019, focusing on nocturnal accumulation, with the findings from Martinez-Eixarch et al. (2018), who conducted a campaign in

2015 using static chambers for approximately 1 to 3 days (10 am to 3 pm) each month across 15 spatially distributed locations.

It would be particularly insightful to include direct comparisons for the specific year of 2015. This year serves as a common reference point for both datasets, allowing for a more precise evaluation of how the RTM monthly averages align with the static chamber measurements from that same year. Highlighting these comparisons could clarify the performance of the RTM in accurately reflecting radon dynamics during 2015.

---

## Author Comment (AC1)

**Answer to Fabian Maier review of *"Estimation of seasonal methane fluxes over a Mediterranean rice paddy area using the Radon Tracer Method (RTM)"* by Curcoll et al.**

*In this study, similarities in the nocturnal mixing between radon and methane ($CH_4$) are used in the so-called Radon Tracer Method (RTM) to estimate the seasonal cycle of $CH_4$ emissions over a rice field area on the Spanish Mediterranean coast. It highlights the potential of the RTM to estimate $CH_4$ fluxes within a limited area, and the importance of $CH_4$ emissions from rice fields during harvest and straw incorporation in the fall season. The latter is a valuable finding for improving $CH_4$ emission inventories.*

Dear Fabian,

First of all, thanks for your review and your positive comments about the importance of the outcomes of this study.

*The manuscript is well structured and written, and provides a thorough analysis and discussion of the RTM approach and its results. I have only minor comments and recommend publication after these have been addressed.*

*Minor comments:*

*You nicely illustrate the performance of the WRF-FLEXPART transport model to simulate radon concentrations (e.g. Fig. 9). These results indicate that the transport model overestimates nocturnal mixing in the boundary layer. I'm wondering how this will affect the RTM results, i.e. the seasonal cycle of the $CH_4$ fluxes. Is there a seasonal cycle in how strong the model underestimates nighttime radon concentrations? Maybe you could briefly discuss to what extent an overestimation of the nocturnal mixing has an impact on the nocturnal RTM footprint and thus on the effective radon flux used to estimate the $CH_4$ emissions.*

Authors want to thank the review for the nice comment. Actually, the possible overestimation of the nocturnal boundary layer of the transport model over the area of interest may affect the effective radon flux calculated for each night within the RTM application. The quantification of this contribution is quite complicated because the radon flux maps themselves have, so far, a not well defined associated uncertainty and mainly over the area of interest.

However, in order to evaluate how the overestimation of the nocturnal boundary layer may affect the simulated footprint utilized within the RTM application, we have analysed here for the reviewer the differences between measured and simulated radon concentrations within the nocturnal temporal window used for RTM. The difference between simulated and measured data have been estimated over the all 2019 for daily nocturnal peaks (Figure AR1).

It can be observed that during spring and summer months the mean values of the observed differences are almost zero or less than 0.5 Bq m$^{-3}$ although a large dispersion is observed due to modelled wind speed, modelled nocturnal PBLH, modelled radon flux, etc. An average difference of around 1.2 Bq m$^{-3}$ is observed in the autumn period. This difference may be attributable to an overestimation of boundary layer height and mixing inside the nocturnal boundary layer for the selected months, but also to an underestimation of radon fluxes from maps. Anyway, both of these two hypotheses may cause an underestimation of the effective radon fluxes, which may cause an underestimation of methane fluxes when applying the RTM. However, although the difference in bias between time periods is significative ($p < 0.05$), the dispersions is high, and it will be difficult to extract conclusions from this limited data. Thus, the following sentence has been added to the text:

*This bias between the observed and modelled radon concentrations at the daily peak was not constant over the tested year. For example, in April and May no bias was observed between radon observations and Flex-WRF-ERA5 modelled radon data. An average bias of 1.21 Bq m$^{-3}$ was found in the months of October-November. This variability may also induce biases in the calculated nocturnal radon fluxes and therefore in the methane fluxes retrieved with the RTM. However, the variability in the bias may not be*

*due only to the calculated radon fluxes but also to the WRF input so it is difficult to quantify and it may be a further analysis.*

[Figure]

**Figure AR1: Differences between observed and simulated (WRF-ERA5) radon concentration at DEC. The green points are the averages for the three selected periods: April-May, July-August, October-November. The green whiskers are the standard deviation of the values.**

*Specific comments:*

*Fig 2.: Does the flooding with sea water affect the entire ERD, and/or is it only temporary? I would not have expected high local radon emissions in December (cf. p. 14, l. 326-328) if the land is flooded. Please briefly describe what flooding with sea water means (can also be done in the caption of Fig. 2).*

The flooding with sea water of the rice fields only affected some zones of the ERD and not all years. A phrase has been added in the text and the figure has been slightly changed.

*"In some years, a flooding with sea water of some of the rice fields was carried out during winter months in order to cope with an ampullaride plague."*

[Figure]

Moreover, the increase in radon concentration may also be attributable to the northwestern winds, predominant in winter, advecting radon when the wind speed is high.

In fact, in order to cope with advected radon and according to other referees' comments, now we are using a 1.5 m/s threshold for wind speed in the selection of available nights for RTM.

*p. 13, l. 305-308: Does this mean that the inventory assumes zero methane emissions for rice fields outside the crop cultivation period? Please clarify.*

No, the inventory takes in consideration the fallow emissions to calculate the average rice field emissions. However, as it is only used to calculate annual emissions, assumes that the methane emission is distributed homogenously during the crop cultivation period. This last two phrases of the paragraph now are as following: *"The inventoried emissions also take in considerations the fallow emissions, although it distributes its emissions among the crop season. Following the IPCC methodology only for the Ebro Delta crop fields, the same value is obtained.".*

*p. 22, l. 401-403: For some events, the model underestimates the measured radon concentrations (e.g. in ~ July, 20). I'm wondering if such biases could be explained by contributions from lateral radon boundary conditions, e.g. if the air masses come from eastern Europe?*

As correctly stated by the reviewer for some events the model underestimates the radon concentrations. During the simulations we only consider for the footprint calculation the 0m-200m layer and this fact may reduce the influence of long range transport on the simulated data. In this study we have used 10-days back trajectories. However, the 4 first days stands for the 95% of the signal, as we have observed that if we were only using 4 days-backtrajectories the difference would be lower than 5 %. As example, here we present a simulated footprint for July 20, 2019 at 05:00 UTC). Left figure shows the footprint doe the total vertical column and right figure only shows it for the lower 200m layer. We observe much more continental influence in the total column than in the 0-200m. Therefore, probably the underestimation on that day could be attributed to a wrong simulation of the vertical transport rather than contributions from lateral conditions.

[Figure]

*Fig. 8: It's quite hard to distinguish the GLDAS & const. curves (at least for color-blind people). Maybe you could use different colors.*

I've slightly changed the figure in order to make it clearer. I've done the same changes in the figures of the Supplement.

[Figure]

*p. 23, l. 418-422: Maybe you want also cite Gerbig et al. (2008) here: Gerbig, C., Körner, S., and Lin, J. C.: Vertical mixing in atmospheric tracer transport models: error characterization and propagation, Atmos. Chem. Phys., 8, 591–602, https://doi.org/10.5194/acp-8-591-2008, 2008.*

Thanks, authors did not know about this study. It has been now added to the list of references.

*Fig. 11: Could you also show the seasonal cycle of the footprint-weighted radon fluxes (in Fig. 11a), which you are using for the RTM? It would be interesting to see whether the radon fluxes used in the RTM are more similar to the very local or to the regional radon fluxes shown in Fig. 11a.*

In agreement with the reviewer requirement we have now added in the plot of Figure 11 the Footprint-weighted radon fluxes and the following paragraph has been added to the manuscript:

*"The footprint-weighted radon fluxes show a different trend than radon flux maps, which may be caused by the seasonality of winds, coming from the northwest in winter months. Overall, the RTM footprint-weighted radon fluxes are similar for both models and in the same order of magnitude as the 70x70km window or the ERD."*

[Figure]

**Figure 11. a) Average annual cycle of [222]Rn exhalation values for the 70 km x 70 km window, ERD and DEC station grid and RTM footprint-weighted for both ERA5 and GLDAS models; b) Bias between observations and modelled radon concentrations for 2019 for the whole day for each of the radon exhalation maps: ERA5 (solid light blue), GLDAS (dashed-dotted light red), constant value (dashed light green) and for the afternoon (15-18h) for each of the radon exhalation maps: ERA5 (solid dark blue), GLDAS (dashed-dotted dark red), constant value (dashed dark green).**

*p. 26, l. 450-451: To assess the reliability of the ERA5 and GLDAS radon flux maps, it might be useful to show here (in Fig. 11b) also the model-data mismatch for afternoon situations only, as you have already shown that the transport model seems to overestimate nocturnal mixing. This could then perhaps allow a better differentiation between deficits in the transport model versus biases in the radon flux maps.*

As suggested by the reviewer afternoon (15h-18h UTC) bias between modelled and measured atmospheric radon concentrations has been calculated and plotted together with the whole day bias in Figure 11b trying to differentiate between deficits in the transport model versus biases in the radon flux maps (please see figure in the point above). Results do not show significant differences which could help to assess the reliability of one radon flux map over the other. This additional analysis will be added to the new version of the manuscript.

*p. 26, l. 455-457: Can you briefly discuss what could cause these larger radon fluxes in December, i.e. which process is not covered by the description of radon exhalation from the soil. This observation could give indications on how to improve the radon flux maps.*

Authors believe that this increase could be driven by the complete drying of rice fields, which is not taken in consideration in the land models.

*p. 29, l. 501-504: In Fig. 5 you show that the CH$_4$ concentrations have a distinct diurnal cycle only between August and November, when the RTM yields elevated CH$_4$ fluxes. Could this finding support your conclusion that, apart from the rice fields, there are no relevant local CH$_4$ emissions, as these would otherwise cause a diurnal cycle in CH$_4$ concentrations, e.g. by accumulation in the nocturnal boundary layer; and that therefore the RTM-based CH$_4$ fluxes describe mainly the emissions from the rice paddies?*

As correctly stated by the reviewer this observation may help to confirm that the methane emissions quantified applying the RTM are mainly due to rice paddies contribution. In order to better explain it for the readers we will now add this sentence within the manuscript:

It can be observed in Figure 5 that the CH$_4$ concentrations have a distinct diurnal cycle only between August and November, when the RTM yields elevated CH$_4$ fluxes. From January to June no methane diurnal cycles are observed at DEC. This result may support the hypothesis that, apart from the rice fields, there are no relevant local CH$_4$ emissions over the DEC RTM footprint. Otherwise, they have caused an evident diurnal cycle in CH$_4$ concentrations, e.g. by accumulation in the nocturnal boundary layer.

*p. 30, l. 518-521: Does the 5.9 kg CH$_4$ ha$^{-1}$ describe the variability of the flux measurements from the different accumulation chambers or is it an estimate for the uncertainty of the annual mean CH$_4$ flux in the ERD, i.e. does it also include the uncertainties of the accumulation chamber method? If the latter is true, I would not call the 5.9 kg CH$_4$ ha$^{-1}$ a "high uncertainty" (it is only 2%). Please clarify.*

After revising the paper where this data has been extracted (see Martinez-Eixarch 2021) and after talking with the authors, we do now know that 5.9 kg CH$_4$ ha$^{-1}$ is only the difference between the estimated fluxes along two years. We have proceeded to remove this uncertainty value, as the uncertainty of the estimated flux should be higher according to the uncertainty of emissions attributed at each of the months.

*p. 31, l. 547-548: The different observation-simulation biases among the months could also be partly due to seasonal differences in the transport model performance (see my first comment).*

In agreement with the reviewer observation, the paragraph has been modified:

*"2) The seasonality observed in the radon exhalation maps from Karstens and Levin (2023) may not be adequately parametrized in the ERD area, as different bias among the months are observed between the modeled and observed atmospheric radon concentrations. Although the biases could also be produced by the seasonal difference in the transport model performance, the estimated radon fluxes at DEC does not seems to take in consideration the seasonality of the water table height within this area."*

*Technical corrections:*

*Throughout: You switched between "backtrajectory" and "back trajectory".*
*p. 8, l. 177: "may be"*
*p. 12, l. 287: "where" (lower case)*
*p. 25, l. 448: delete "it"*
*p. 28, l. 477: "WRF-GLDAS"*

They have been corrected, thanks

*Supplements:*

*Fig. S2: Is the map shown in panels d-f the 70 km x 70 km window or rather the 150 km x 150 km window? It appears that you are referring to this window as the 150 km x 150 km window in Fig. 3 in the manuscript.*

You are, right, it has been now clarified.

*Fig. S3: If you want, you could also mark these synoptic situations in the time series plots in Fig. S5. Then one could directly see the model-data mismatch associated with these synoptic situations. Typo in the caption: " … the logarithm … "*

Verticals red lines are now marked in Figure S5 corresponding to the simulated dates in Figure S3.

---

## Author Comment (AC2)

***Answer to Scott Chamber review of "Estimation of seasonal methane fluxes over a Mediterranean rice paddy area using the Radon Tracer Method (RTM)" by Curcoll et al.***

*This paper seeks to estimate the methane flux, and its seasonal variability, from a particular rice paddy region on a coastal river delta, using the "local scale" nocturnal-accumulation implementation of the radon tracer method (RTM). The writing is clear and the manuscript well structured. As outlined in the introduction, characterisation of agricultural methane emissions is an important step toward ratifying the Paris agreement.*

*Despite the importance of the intended goal, I cannot recommend publication of this manuscript in its current form because I am not convinced that the RTM, in the way it has been applied, is an appropriate approach to achieve the stated project goals (i.e., targeting the specific rice paddy area). While seasonal methane fluxes are indeed derived, I believe it is unlikely that they are truly representative of only (or predominantly) the ~200 km² of rice paddies in this region. I outline my concerns in more detail below.*

***General comments***

*The radon tracer method as intended to be applied in this study treats the nocturnal boundary layer as a simple box and seeks to use changes in concentration of radon and a companion species (methane, $CH_4$) within this box over a collection of single nights, along with simulated radon emission rates from the bottom surface of this box, to infer the flux of the companion species each night over the same spatial region.*

*There are some necessary assumptions for this technique to be applied:*

*1. Concentration changes within this box should be a function of only the average flux magnitudes over the base of the box (i.e., local flux contributions), and changes in height of the box.*

*2. For advective effects to be ignored, the flux of each gas should be similarly distributed and homogeneous across the base of this box.*

***Regarding assumption 1:***

*It has been well established (e.g., Sesana et al 2003; Chambers et al 2015) that significant accumulation of trace gases in the nocturnal boundary layer, driven specifically by local sources, only occurs when wind speeds are less than about 1.5 m.s⁻¹. At higher wind speeds, observed concentration changes at a given location become increasingly dominated by advection from non-local sources. Under such conditions, confusion can arise between the "local scale" nocturnal accumulation implementation of the RTM as described by Levin et al (1999, 2011), and the "regional scale" implementation of the RTM as described by Biraud et al (2000). According to Figure S4 b, only a relatively small fraction of wind speeds in the nocturnal windows used for the RTM in this study are below 2 m.s⁻¹. Given the 6-hour nocturnal window applied here for the RTM, and maximum acceptable wind speed of < 2 m.s⁻¹ to observe a local influence, the length of the "box" contributing to the observed flux signal would be around 43 km (if only the low wind speeds of this study were considered). The rice paddy region being investigated (based on Fig S1) appears to have dimensions of only around 20 km (east-west) x 12 km (north-south). Based on the location of the measurement site, the rice paddy field fetch in the dominant wind directions (NW and SSE) is only 2 – 10 km. So, even in the most ideal, low wind scenarios, the contribution of the rice paddy area to the overall signal observed is a low fraction.*

***Regarding assumption 2:***

*Given that the best-case scenario base length of the idealised "box" from the measurement point is around 43 km, and (a) the site is coastal, and (b) the rice paddy fields are of limited spatial extent, and periodically flooded, neither fluxes of radon or $CH_4$ over this distance (in any direction from the measurement site) would approximately homogeneous or similarly distributed.*

*If only stable nocturnal conditions with wind speeds less than or equal to 0.5 m.s⁻¹ were targeted, this would limit the local fetch contribution over the 6-hour nocturnal accumulation window to around 10 km, which*

*may work for wind directions west through south, but this would severely limit the amount of data available for analysis.*

First of all, we want to thank Scott Chambers for his comments.

The improvement of the Radon Tracer method (RTM) for its application to indirectly estimate greenhouse gases flux is nowadays a theme of interest within the scientific community. In fact, the Integrated Carbon Observation System (ICOS) is already including radon measurements in Europe and its Carbon Portal is going to offer GHG RTM based fluxes as products. For these reasons, as Scott will certainly know as part of it, an European project (EMPIR project 19ENV01 traceRadon, Budget: 2 M€) was run between 2022 and 2024 to improve the radon concentration and radon flux metrology and to generate first guidelines for RTM applications. This is due to the fact that the RTM has not yet a well-established protocol to be applied and studies of this type are highly relevant not only for meeting the goals of the Paris Agreement but also for testing the RTM in extreme environments.

It is true and already known (Grossi et al., 2018) that the two main assumptions for the correct application of this method are the ones cited previously by Scott. In the following lines we answer to these two main points and we evaluated how this may impact the results of our study. We do seriously think that this study is worth of being published as well as it has been also expressed by the other two reviewers of this manuscript. The RTM, so far, is proposed and investigated by the scientific community as another tool to retrieve greenhouse gases fluxes over the area. The RTM is presented as a complementary method, cost efficient, which wants to support results obtained using intense experimental accumulation chamber campaigns or inverse modelling techniques. It is important to underline as the results of these study related with methane fluxes over the rice paddy area, both in its seasonal variability as well as in its absolute mean values, match coherently with results obtained by other researchers using expansive accumulation chamber measurements campaigns over the same area. Furthermore, this study clearly shows the miss of seasonality and information of the available inventories such as the EDGAR.

Reviewer may consider that the main aims of this study were:

1. Comparing RTM based results obtained at DEC station during different phases of the rice production cycle with other experimental studies carried out in the same area (Martínez-Eixarch et al., 2018, 2021);

2. Analysing the seasonal variability of the methane fluxes over the area mainly due to the rice production and offer new information to improve static GHG inventories, such as EDGAR v7.0;

3. Using atmospheric radon concentration measured at the station to test the application of traceRadon radon flux maps, based on different soil moisture dataset, for radon concentration simulations at a sea-soil border area such as Ebro river delta.

All these previous objectives have been fully accomplished within the study.

Finally, we have observed that all specific comments done by the reviewer are basically related with lines of the manuscript about the previously cited assumption 1 and 2. Thus, we will not specify them point by point because they will be corrected within the new version of the manuscript to better clarify these aspects in agreement with our answers.

**Regarding assumption 1:**

In order to analyse this possible limitation in our study, an extra analysis has been carried out running the RTM only over nights when wind speeds are lower than 1.5 m/s within the RTM application. Methane fluxes calculated with the RTM applying this previous restriction are compared with methane fluxes when no wind restriction is applied in the new **Figure 12**, which will be now included in the new manuscript version. It can be observed that the methane fluxes variability does not really change when wind speeds restrictions are applied. Methane fluxes mean values differences are observed only during the month of September. However, when the wind speed criteria are applied, the number of available nocturnal events is lower and no events were available for February which is typically characterized by strong winds coming to the station (Section of Figure 4 from Grossi et al., 2016 here reported). For this reason, we have decided

that the calculation of the annual methane fluxes was carried out without wind speed criteria application for the robustness of the analysis.

***Regarding assumption 2:***

The small effect observed at Delta Ebro area when the wind speed limitation is applied is probably due to:

- Despite the fact that the rice paddy extension is limited, its methane emissions are incredibly high so it is basically covering other unknown methane sources over the whole Delta Ebro river area. Rice paddies represent in this case a hot spot over the Delta Ebro area where not more significant methane emissions are present.

-The DEC station has a really low tower where the air sample intake is occurring (10 m. a.g.l). This, together with the only 6 hours back trajectories used within the Flexpart-WRF model, unsure the reduced size of the DEC station footprint. This is not comparable with ICOS tall towers which may have much bigger areas of influence. Results of our study show that the less of 25% of DEC footprint signal has continental origins. When the 1.5 m/s threshold is applied, the continental influence is reduced to 15%.

[Figure]

**Figure 12. Left: Boxplot of the methane RTM based fluxes over the DEC station calculated using Flex-WRF-ERA5 (blue) and Flex-WRF-GLDAS (red) for every month of the year (period 2013-2019) for the whole dataset (left) and only takin in consideration events with ws<1.5 m/s. Outliers are represented with round points, boxes represent the region between interquartiles Q1 and Q3, and horizontal solid lines the medians for each model.**

[Figure]

**Section of Figure 4 from Grossi et al., 2016.**

*Specific comments*

*L44: The RTM requires that both gas source functions are similarly distributed over the region in question. For the ~5 months of the year that the rice paddies are flooded, there is little impediment to $CH_4$ emission (via diffusion, ebullition, and transport through aerenchyma). This is not the case for radon emission. When the soil profile is flooded, radon generated in the saturated soil matrix below the soil surface is unlikely to make it into the atmosphere before decaying. The much smaller fraction of radon generated at or near the soil surface is not produced in sufficient quantities for bubble transport, so diffusion is its only pathway (which would likely take of order 2-3 days to get through the 8 – 15 cm of standing water). Presumably this flux would only be marginally higher than typical open water radon fluxes, and therefore much lower than anything predicted by the European flux maps. Beyond the edges of the rice paddies there is the opposite problem. Radon fluxes would be higher from the unsaturated soils, and $CH_4$ emissions lower.*

Thank you, again the reviewer for the observation.

First of all, we would like to undelight that there are several studies that show that radon is also released (in small quantities) in shallow waters, and that it exists a transport of gases through the stem of vegetation which increases the release of the radon fluxes (Kozak et al., 2003; Megonigal et al., 2020). Therefore, even when rice fields are partially flooded during the productive rice months, there could be enough radon release from fields to be detectable. The nocturnal accumulation when the wind is nearby zero supports this affirmation.

Actually, looking again at Section of Figure 4 from Grossi et al., 2016 it can be observed that for the most of the season, mainly in summer and autumn low-medium winds are present at DEC station. In these two previous seasons, as reported by Grossi et al., 2016 (Figure 7, central column here pasted), it can be observed that nocturnal radon concentration increases at DEC station up to 6.5 Bq m$^{-3}$. This fact may support the hypothesis of not negligible radon fluxes over this area also when rice paddies area is flooded.

In addition, authors think that it is important to clarify that, as reported within the manuscript, the effective radon flux seen by the station was calculated not only using the radon flux map but coupling them with the Flexpart based footprints to weight the total area contribution as explained in Grossi et al., 2018 (Equation 3).

[Figure]

**Section of Figure 7 from Grossi et al., 2016.**

*L72: Clearly the modelled radon fluxes for this region do not represent completely inundated conditions (which is the case for the region of interest for 5 months of the year). Presumably they are representing seasonal changes in soil moisture for uncultivated land with a particular Ra-226 content. [I noticed later that this is acknowledged in Figure 11a – but I assume that the modelled fluxes were still used for this study, which would not correctly represent the study region].*

We thank the reviewer for the observation.

Currently, the available radon flux maps from the traceRadon project used in this study are not free from uncertainties or bias. Just taking in consideration the difference between the two models ERA5 and GLDAS, for some regions as central Europe there are huge biases between both models, sometimes above the 100% of the estimated radon flux. One of these uncertainties may be the representability of the seasonality of radon fluxes. This, unlikely, also happen with GHG emission inventories such EDGAR or biogenic flux models. Despite the fact of not existing yet a perfect radon flux map, the utility of these maps is here shown because they allow a good estimation of the seasonality of methane fluxes due to the anthropogenic activity which perfectly match with experimental accumulation chamber measurements from other authors over the same area. The objective of the paper is not to overcome with a quantitative annual flux but to show the possibility of the application of the RTM to study methane fluxes in a region such as the Ebro delta to support agriculture techniques and GHG emission reduction actions. However, we agree that the limitation of the study, because of reliability of radon flux seasonality over this area, may be underlined in the discussion and conclusion of the manuscript.

*Fig 2: In the month of land preparation (and period of straw incorporation), is an increase in local radon flux expected due to the tilling (and associated increase in porosity / exposed surface area)? According to Figure 4 observed radon concentrations peak at these times. The authors might check back trajectories to see whether there is a notable difference in airmass time over land for these periods, or whether there might be a local change in radon flux.*

Authors thank the reviewer for his suggestion but a detailed analysis of radon flux in each moment of the soil treatment is far from the aim of this study. However, it can be an interesting further study by experimental radon/methane flux campaigns.

*Section 2.2: the station is not situated well for RTM observations. Fetch in the prevailing wind direction (NW) is limited to around 2 km, wind from the longest fetch region (W) is uncommon, and immediately to the S-SE of the site is a large body of standing water (before the rice paddies continue). Even when the fields are not inundated, this setting would yield large spatial gradients in radon flux. Furthermore, for the period when the fields were inundated, if measurements \*were\* targeting just the rice paddy fields, the radon flux from this region (within typical measurement error) would be essentially zero.*

Authors thank the reviewer for the observation. We know that the site is an extreme site, not only for being a costal site but also because it is located into a Delta. This one of the many reasons because this study is really interesting and worth to be shared with the scientific community.

As we explained previously, the DEC station inlet is only at 10 m above the ground level. This fact heavily reduces the footprint of the area. Actually, DEC was not included within the INGOS network because of its small footprint and local signal. Prove of this is the fact that when we apply a wind speed limitation the results, especially in seasonality, do not vary. The radon seen by the station, as well as the methane, is due to the radon/methane emitted over the footprint area of the station. This effective radon flux was calculated coupling the radon flux maps with the Flexpart footprint following the methodology explained in Grossi et al., 2018. Finally, despite the fact that radon flux may be low over this area, the nocturnal accumulation of radon activity concentrations measured over the 7 years dataset and shown in this study and by Grossi et al., 2016 justify that this exhalation is not zero. All this will better explain within the discussion section of the revised manuscript.

*L165-166: The authors claim that the RTM here is applied over the footprint of the study area. The study area (200 km²) measures roughly 20 x 12 km in dimensions. According to Fig S4b, wind speeds in the nocturnal RTM window reach as high as 22 m.s⁻¹, which would cover a distance of 475 km over the 6-hour nocturnal window. Even the 'mid-range' nocturnal wind speed used of 5 m.s⁻¹ would cover a distance of almost 110 km in this time. Based on these values, the CH₄ flux signal retrieved by this method from the actual intended study region would only constitute a small fraction of the result (i.e., observations would be dominated by advection from non-local regions).*

*L170-171: For wind speeds well over 2 m.s⁻¹, and large spatial variability in the radon flux in the vicinity of the measurement site, it is not possible to make the assumption of negligible advection effects.*

*L176-177: The "local scale" nocturnal accumulation method of RTM application to derive local fluxes only make sense under conditions of nocturnal stability (not when the nocturnal atmosphere is near-neutral or well mixed). If nocturnal stability criteria are included in the selection of appropriate conditions under which to apply the RTM, these are usually also an effective filter of rainfall events. Meaning that, over the 6-hour nocturnal window, there would be less chance of rainfall events, and less chance of variable radon fluxes over the nocturnal window period each night.*

*L188: The challenge here is knowing the footprint area represented by the measurements, and how this relates to the actual region of interest.*

All these previous comments related with manuscript sentences are related with the reviewer first observation about assumption 1 of RTM. We think that the new analysis using wind speed lower than 1.5 m/s helps to explain this better and it will be included into the new version of the manuscript.

*L190-191: Why is an influence area of 70 km x 70 km used, when the actual dimensions of the study area are 20 km x 12 km? Application of equation (3) assumes similar distribution of both fluxes over the region of interest, and homogeneity of the respective fluxes over this region, which is clearly not the case when air masses leave the rice fields to the west or cross the coast to the ocean in any other direction. Based on Fig S3 f, if the boxes represent hourly intervals, this still indicates a contributing fetch region over the 6-hour nocturnal window that is over 3 times the scale of the intended measurement region. Calculating an average radon flux over a region of that size and heterogeneity, will not result in a value representative of the intended study region.*

In the figure 10 the average footprint for the RTM nights is plotted. In line 449 we explain that the continental influence is only the 25% according to footprints. Taking in consideration that the models overestimate the nocturnal wind speed, the influence should be much lower. We have recalculated the footprint influence area map in the case of only selecting events with ws <1.5 m/s. It is plotted in the following figure which will be added to the manuscript. When we only select events with wind speed lower than 1.5 m/s, the continental influence decreases to 15% according to models, although it is probably lower due to the overestimation of nocturnal winds of the mesoscale models.

[Figure]

*L200-206: No wind speed or stability criterion are used in the selection of nights on which to apply the RTM, when stable conditions are in fact the most necessary criterion for applying the nocturnal accumulation form of the RTM to retrieve local scale fluxes. Also, somewhat dangerously, a strong linear correlation between radon and CH4 is not – on its own – a reliable indicator of a time representing a good local flux measurement (this approach is more like that applied by Biraud et al., 2000 for regional RTM flux estimates). Such conditions can arise under strong advection (high winds) and be completely unrelated to the local flux. Similar arguments can be made regarding positive concentration gradients if they are taken in isolation (e.g., without also considering the nocturnal stability state).*

Thanks Scott, we think that this point is also related with your first observation about assumption 1 of RTM. The results of the new analysis (calm conditions) help to explain this better and it will be included into the manuscript.

*L209-212: A problem with using models to estimate footprint areas for RTM calculations is that stable nocturnal conditions are a requirement for applying the nocturnal accumulation version of the RTM, and these are the conditions under which models have the poorest performance. Mixing depths, wind speeds and footprint regions tend to be significantly exaggerated.*

Authors thank the reviewer for his comment. We also know that that models do not perform very well in simulating the nocturnal boundary layer and the surface wind speed. We know it's a source of uncertainty and so is commented on the text.

Related to this, we know that this overestimation of mixing heights and wind speeds is causing an underestimation of the radon nocturnal peak. In the answer to a second reviewer , we have analysed the variability of this underestimation and it will be added into the final version of the manuscript.

It can be observed that during spring and summer months the mean values of the observed differences are almost zero or less than 0.5 Bq m$^{-3}$  although a large dispersion is observed due to modelled wind speed, modelled nocturnal PBLH, modelled radon flux, etc. An average difference of around 1.2 Bq m$^{-3}$ is observed in the autumn period. This difference may be attributable to an overestimation of boundary layer height and mixing inside the nocturnal boundary layer for the selected months, but also to an underestimation of radon fluxes from maps. Anyway, both of these two hypotheses may cause an underestimation of the effective radon fluxes, which may cause an underestimation of methane fluxes when applying the RTM. However, although the difference in bias between time periods is significative

(p < 0.05), the dispersions is high, and it will be difficult to extract conclusions from this limited data. Thus, the following sentence has been added to the text:

*This bias between the observed and modelled radon concentrations at the daily peak was not constant over the tested year. For example, in April and May no bias was observed between radon observations and Flex-WRF-ERA5 modelled radon data. An average bias of 1.21 Bq $m^{-3}$ was found in the months of October-November. This variability may also induce biases in the calculated nocturnal radon fluxes and therefore in the methane fluxes retrieved with the RTM. However, the variability in the bias may not be due only to the calculated radon fluxes but also to the WRF input so it is difficult to quantify and it may be a further analysis.*

[Figure]

**Figure AR1: Differences between observed and simulated (WRF-ERA5) radon concentration at DEC. The green points are the averages for the three selected periods: April-May, July-August, October-November. The green whiskers are the standard deviation of the values.**

*L213-221: If eqn (3) is being used to derive $CH_4$ fluxes from the rice paddy region, then only radon fluxes representative of this region are meaningful, and conditions need to be selected to avoid the measurement footprint significantly exceeding the bounds of the study region. Do the radon flux maps account for the complete inundation of the rice fields for almost half of the year? [I now see Figure 11a confirms that they don't]*

*L257-260: This implies that a representative radon flux for the 20 km x 12 km study region is being derived based on a land fetch of ~60 km or more. Presumably most of this land is not inundated for 5 months of the year? Also, if radon fluxes from this fetch are contributing to the observations, doesn't this also apply to the $CH_4$ fluxes? (which does not match the study goals)*

Thanks Scott, we think that this point is related with the comment you did at the beginning of the Specific comments section so we do not need to answer again. The paragraph will be changed within the text.

*L328: Based on the wind speeds at this site (and daytime minimum radon concentrations in December, Fig 5), the higher December observed average radon concentration would most likely be fetch related, rather than due to a sudden change in local radon flux for a single month.*

As reported in the previous plots from Grossi et al., 2016 it can be observed that in December, and the rest of the winter. However, only in December we are observing this radon increase, which make us think about an increase of the local source, probably coupled with the advected radon from the north-west. This observation is repeated along the 7 years dataset.

L354: Northwestern

Corrected, thanks

Fig S4 b (and Fig 7c): Most likely any RTM results derived for nocturnal wind speeds > 2 m·s⁻¹ will not closely represent what is actually happening over the rice paddies. This appears to be the case for a large fraction of the dataset.

Thanks Scott, we think that this point is related with the comment you did at the beginning of the Specific comments section so we do not need to answer again. The paragraph will be changed within the text.

L364-365: As previously mentioned, the overestimation of wind speeds by the model at night will lead to exaggerated footprint estimates.

Thanks Scott, we think that this point is related with the comment you did at the beginning of the Specific comments section so we do not need to answer again. The paragraph will be changed within the text.

L429-430: At night, under near-neutral or stable conditions, a measurement height of 10 m a.g.l., is not a guarantee of a "very local" fetch when wind speeds exceed 1.5-2 m.s⁻¹.

Thanks Scott, we think that this point is related with the comment you did at the beginning of the Specific comments section so we do not need to answer again. The paragraph will be changed within the text.

L432-433: Considering the dominant wind directions within the nocturnal window for RDM application (NW or SSE), it would be uncommon for air masses of a given event to spend more than 15 – 20% of their time over the intended study region. I don't believe that this is selective enough to achieve the study goal.

Thanks Scott, we think that this point is related with the comment you did at the beginning of the Specific comments section so we do not need to answer again. The paragraph will be changed within the text.

L433-434: There are a lot of coastal areas in the oceanic fetch region when the wind direction is from the NW – is it not the case that coastal regions can be sources of methane? Of course, coastal oceanic radon fluxes are also higher than open ocean radon fluxes, but they are still MUCH lower than any terrestrial radon fluxes.

Thanks Scott. Yes, oceanic methane emissions exist (Capone and Hutchins, 2013; Egger et al., 2018), and are mainly driven by shallow areas (Weber et al., 2019). However, even taking in consideration the highest possible emissions for an area such as the Mediterranean (1 mmol m⁻² y⁻¹, according to Weber et al., 2019), it is about 3 orders of magnitude lower than the methane flux accounted in the RTM for the ERD. Therefore, we can consider it negligible. Furthermore, our results are coherent with results obtained using methane accumulation chamber only over the rice paddies.

*L455-457: Use 10-day back trajectories and calculate average time over land for air masses within the ABL (or residual layer) – i.e., below ~2000 m – and compare this information with the monthly radon plot of Figure 4. If they show similar trends, then fetch effects are a greater influence on the observed radon concentration than seasonal changes in the local radon flux.*

Thanks for the idea Scott. This analysis so far is behind the scope of our study but we will take into account your suggestion for future analysis of the radon behaviour in extreme coastal areas.

*L501-503: Based on Fig S3, even air masses arriving at the measurement site from the ocean (NE) could have been over other land regions within the last 1-4 days. This means, they could likely still have significant correlated events of radon and CH₄ from prior land contact at the time they cross the local coast of the measurement site. So, it is not safe to assume that the Rn and CH₄ content of an airmass from the coast, that then crosses a limited extent of rice paddies, is only representative of exchanges from the rice paddy region.*

Thanks for the comment. We do not say it is "only representative", as it could not be said of any other RTM analysis looking for local fluxes. In our work we say that "mainly represents" or "on average, it represents a proportion higher than 75%", according to the footprint study. The correlation with fluxes obtained with accumulation chambers confirms our results.

*L530-532: Under nocturnal conditions, with moderate to high wind speeds, the assumption of the measurement fetch being limited to a few km is not valid.*

Thanks Scott, we think that this point is related with the comment you did at the beginning of the Specific comments section so we do not need to answer again. The paragraph will be changed within the text.

**References**

*Biraud and co-authors: European greenhouse gas emissions estimated from continuous atmospheric measurements and radon 222 at Mace Head, Ireland, JGR. Atmos., 105(D1), 1351–1366, doi:10.1029/1999JD900821, 2000.*

*Chambers and co-authors, 2015. On the use of radon for quantifying the effects of atmospheric stability on urban emissions. Atmos. Chem. Phys. 15, 1175-1190.*

*Levin and co-authors: Verification of German methane emission inventories and their recent changes based on atmospheric observations, JGR. Atmos., 104(D3), 3447–3456, doi:10.1029/1998JD100064, 1999.*

*Levin and co-authors: Verification of greenhouse gas emission reductions: the prospect of atmospheric monitoring in polluted areas, Philos. Trans. R. Soc. A Math. Phys. Eng. Sci., 730 369(1943), 1906–1924, doi:10.1098/rsta.2010.0249, 2011.*

*Sesana, L., Caprioli, E., and Marcazzan, G. M.: Long period study of outdoor radon concentration in Milan and correlation between its temporal variations and dispersion properties of atmosphere, J. Environ. Radioactiv., 65, 147–160, doi:10.1016/S0265-931X(02)00093-0, 2003.*

References

Capone, D. G. and Hutchins, D. A.: Microbial biogeochemistry of coastal upwelling regimes in a changing ocean, Nat. Geosci., 6(9), 711–717, doi:10.1038/ngeo1916, 2013.

Egger, M., Riedinger, N., Mogollón, J. M. and Jørgensen, B. B.: Global diffusive fluxes of methane in marine sediments, Nat. Geosci., 11(6), 421–425, doi:10.1038/s41561-018-0122-8, 2018.

Kozak, J. A., Reeves, H. W. and Lewis, B. A.: Modeling radium and radon transport through soil and vegetation, J. Contam. Hydrol., 66(3–4), 179–200, doi:10.1016/S0169-7722(03)00032-9, 2003.

Megonigal, J. P., Brewer, P. E. and Knee, K. L.: Radon as a natural tracer of gas transport through trees, New Phytol., 225(4), 1470–1475, doi:10.1111/nph.16292, 2020.

Weber, T., Wiseman, N. A. and Kock, A.: Global ocean methane emissions dominated by shallow coastal waters, Nat. Commun., 10(1), 1–10, doi:10.1038/s41467-019-12541-7, 2019.

---

## Author Comment (AC3)

**Answer to Dafina Kikaj review of Curcoll et al. (2024), "Estimation of seasonal methane fluxes over a Mediterranean rice paddy area using the Radon Tracer Method (RTM)"**

*This paper aims to estimate methane flux and its seasonal variability using the "local scale" Radon Tracer Method (RTM) in a rice paddy in the Ebro Delta. The manuscript is well-written and logically structured, with a clear presentation of methodology and results. Given the importance of methane emissions from rice paddies as a significant contributor to agricultural greenhouse gases, as highlighted in the introduction, this study is highly relevant for meeting the goals of the Paris Agreement.*

*However, before recommending publication, I would suggest further clarification on some major points.*

The authors want to thank Dr Dafina Kikaj for her review. We appreciate all her comments and we have tried to address them in the following lines

*Methodology:*

*The "near flat variability of radon" raises significant concerns in this study. The authors are conducting measurements at a site that remains flooded for much of the year, which would lead to no radon flux. In contrast, methane ($CH_4$) emissions may still occur from this area. Given that this is a coastal site, radon flux from the ocean is minimal, but coastal regions can still generate $CH_4$.*

*The local accumulation version of the Radon Tracer Method (RTM) assumes that the fluxes of both radon and methane are similarly distributed in space and homogeneously spread across the measurement footprint. This homogeneity is crucial for neglecting advective effects. Additionally, the "accumulation" model is valid only under very stable conditions, specifically when wind speeds near the surface are less than or equal to 1.5 m/s. If the wind speeds exceed this threshold, the correlations observed between radon and $CH_4$ would likely result from fetch effects rather than local accumulation (This limitation is also addressed in greater detail by another reviewer, Scott Chambers.).*

*To achieve their objectives more effectively, the authors would need to focus exclusively on low wind speed nights, which, based on their wind rose data, are quite rare. This restriction would significantly reduce their dataset. Furthermore, they could only apply their methodology during times when the rice paddies were not flooded, ensuring that radon flux was indeed present from the relevant fetch region.*

Authors thank the reviewer for underlying this possible limitation of the study.

Actually, looking at the Figure 7 of the manuscript (panel c: observations over the nocturnal window (23h UTC to 03h UTC), it can be observed that weak winds (below 2 m/s) are observed a DEC station during 5%-7% of the nights. Grossi et al., 2016 (Figure 4 here pasted) shows that, depending on the season of the year, these weak nocturnal winds may occur more frequently (up to 10%). In summer and autumn for example. In these two previous seasons, as reported by Grossi et al., 2016 (Figure 7, central column here pasted), it can be observed that nocturnal radon concentration increases at DEC station up to 6.5 Bq m-3.

There are several studies that show that radon is also released (in small quantities) in shallow waters, and that it exists a transport of gases through the stem of vegetation which increases the release of the radon fluxes (Kozak et al., 2003; Megonigal et al., 2020). Therefore, even when rice fields are partially flooded during the productive rice months, there could be enough radon release from fields to be detectable. The nocturnal accumulation when the wind is nearby zero supports this affirmation.

[Figure]

**Section of Figure 4 from Grossi et al., 2016.**

[Figure]

**Section of Figure 7 from Grossi et al., 2016.**

However, in order to analyse this possible limitation in our study, and in agreement with Dafina Kikaj and Scott Chambers suggestions, we have now included the restriction for events with wind speed lower than 1.5 m/s within the RTM application. Methane fluxes calculated with the RTM applying this previous restriction are compared with methane fluxes when no wind restriction is applied. The following figure, which will be now included in the new manuscript version, shows the methane fluxes when the wind criteria restriction is applied and when no wind restriction is applied. It can be observed that the methane fluxes variability over the months continues to be the same. Methane fluxes mean values differences are observed only during the month of September. Obviously when wind speed criteria is applied the number of available nocturnal events is lower (45%of them) and no events were available for February which is typically characterized by strong winds coming to the station (Section of Figure 4 from Grossi et al., 2016) . For this reason, the calculation of the annual methane fluxes was carried out without wind speed criteria application.

[Figure]

**Figure 12. Boxplot of the methane RTM based fluxes over the DEC station calculated using Flex-WRF-ERA5 (blue) and Flex-WRF-GLDAS (red) for every month of the year (period 2013-2019) for the whole dataset (left panel) and for the filtered dataset when wind speed measured at DEC is < 1.5 m/s. Outliers are represented with round points, boxes represent the region between interquartiles Q1 and Q3, and horizontal solid lines the medians for each model.**

***More detailed points in methodology:***

*The dataset spans from 2013 to 2019, providing approximately seven years of data with a total of 61,320 hourly measurements. I would appreciate some clarification on the rationale for specifically including data from 2019.*

The RTM analysis was applied to the full 7 years' dataset using, for the calculation of the effective radon fluxes seen by the station, the footprints calculated during the all selected night.

However, the year 2019 was selected specifically to perform the WRF and Flexpart models evaluation.

*However, only about 30% of this data (approximately 18,396 observations) was usable due to instrument maintenance issues. Additionally, limiting the analysis to just six hours per night further restricts the dataset, raising concerns about the representativeness of the flux measurements. For example, Table 2 shows that data from February across seven years includes usable measurements from only two nights.*

Authors agree with the reviewer that unfortunately due to the extreme climate conditions of the DEC station over the 7 years only the 30% of the dataset was available for the RTM analysis. In addition, the RTM has to be applied during calm nights and this decrease the number of events. The limitations of the RTM method were already presented by other authors (Grossi et al., 2018; Levin et al., 2021) and they are also addressed in this manuscript.

However, the RTM is not here presented us only possible method to estimate GHG fluxes but as a reliable independent tool which could be useful for supporting other methodologies (experimental accumulation chambers campaigns, inverse modelling, etc.).

Reviewers should note that it was out of the aim of this study analysing the daily variability of methane fluxes over the area, due to the RTM time window application, but authors were mainly interested in:

1. Comparing RTM based results obtained at DEC station during different phases of the rice production cycle with other experimental studies carried out in the same area (Martínez-Eixarch et al., 2018, 2021);

2. Analysing the seasonal variability of the methane fluxes over the area mainly due to the rice production and offer new information to improve static GHG inventories, such as EDGAR v7.0;

3. Using atmospheric radon concentration measured at the station to test the application of traceRadon radon flux maps, based on different soil moisture dataset, for radon concentration simulations at a sea-soil border area such as Ebro river delta.

All these previous objectives have been fully accomplished without the study. In addition, it is important to underline that despite of the dimension of the dataset applied for the RTM analysis, the most interesting months to evaluate the methane due to rice paddy have enough radon/methane concentration data to ensure a good representability.

*The diurnal variability of radon appears very flat, which may be attributed to the site's proximity to the coast and the limited radon fluxes present.*

As show in Figure 5 of the manuscript daily radon variability is in the order of 2-3 Bq m$^{-3}$ between nocturnal and daily hours. This is most probably due, as the reviewer said, to the low radon fluxes over a coastal area. However, in the previous version of this plot the common y-axis limits between the December radon concentrations and the others months did not help. The graph has been now modified in order to show a clear scale and better represents the diurnal cycle observed in all months.

*Finally, the data selection criterion employed a threshold of $R^2 \geq 0.5$, which is relatively low; a minimum of 0.7 is generally preferred. This suggests that the observed diurnal radon signal may arise from air advected to the study site or from minimal contributions from exposed ground. During other times of the year, wind speeds are often quite high. When comparing this distance to the scale of the rice paddy fetch in various directions from the measurement site, the implications for flux representativeness become even more pronounced.*

The application of RTM has not yet been harmonized and there is not yet a common set of criteria to be used. For example the criteria of using $R^2 \geq 0.5$ has been already used by other authors such as Levin et al. (1999), Hammer and Levin (2009) and Wada et al (2013). Thus, authors decided to use this criteria in order to have a methodology applied coherently with past studies. Furthermore, the application of low wind speed criteria (<1,5 m/s) may help to reduce advection events within the analysis.

*I understand that you are using the Weather Research and Forecasting (WRF) model, which operates on a mesoscale, to estimate the influenced footprints (for a local). However, in section 3.2.1, regarding the evaluation of the meteorological model, it's noted that the correlation between simulated wind speed and observed wind speed is 0.57. Additionally, it appears that the model tends to overestimate wind speeds for most of the assessment period.*

*Given these findings, it is important to critically evaluate the accuracy of the model in estimating the footprint dimensions. Overestimating wind speed can lead to significant discrepancies in how gases are predicted to disperse. If the model is consistently overpredicting wind speeds, the resultant footprints may be smaller or inaccurately positioned, which could misrepresent the true spatial extent of influence.*

*Furthermore, it is essential to clarify whether this correlation of 0.57 applies uniformly across the entire study period from 2013 to 2019, or if it is specific to certain times of day, such as nighttime or daytime. Variations in wind patterns between day and night can significantly affect dispersion characteristics. If the correlation is weaker during certain periods, this could further impact the model's reliability in estimating footprint dimensions.*

*To improve the robustness of your analysis, consider the following:*

1. ***Temporal Analysis****: Investigate whether the correlation varies by time of day. This can provide insights into how well the model performs under different meteorological conditions.*

2. ***Sensitivity to Overestimation****: Assess how the overestimated wind speeds affect gases and footprint dimensions. This sensitivity analysis can help quantify the potential impacts of the model's inaccuracies.*

The WRF model output were evaluated using experimental data measured at DEC station only over the period 2019. Results obtained are coherent with literature studies. Actually, models are known to generally overestimate wind speed at surface and mainly at night. When comparing the correlation or RMSE results from the meteorological fields obtained in our study, they do not differ from other studies carried out in coastal areas (Cerralbo et al., 2015; Takeyama et al., 2013).

1. As suggested by the reviewer a temporal analysis was realized of the correlation between modelled and observed values and it will be included in the supplement material of the revision version of the manuscript.

In the following figure, the RMSE, the bias and the correlation between modelled and observed wind speed is shown over the seasons for the all day or only for the nocturnal RTM window. No significative differences are observed between results when we only consider the RTM window and differences when considering the whole day. RMSE is similar for all the 2019 except for November, when a huge bias produces an increase in the RMSE. This huge bias in November is produced by a high overestimation of the nocturnal wind speed for this month. This phenomena, however, produces a better correlation for wind speed for th<t month.

[Figure]

2. Concerning the effect of the wind speed overestimation on the RTM suggested by the reviewer, we have investigated it and we have now update Figure 11a including also the flux derived from the RTM footprint. The peak observed both for ERA and GLDAS RTM footprints in November is probably due to the overestimation of nocturnal winds from the north-west in these months as seen in the previous Figure. Therefore, in November, an excess of continental contribution is probably modelled. Real radon fluxes would therefore be lower for that months, meaning that probably the methane fluxes are also lightly overestimated in November which is coherent with the results observed in Figure 12.

This new analysis will be included in the new version of the manuscript.

[Figure]

**More specific comments:**

**L328:** *I remain unconvinced that the average radon levels observed in December can be attributed solely to local radon fluxes. The high standard deviation associated with this mean suggests significant variability in the data, indicating that the observed radon levels may be more representative of a unique, one-time event rather than a consistent, ongoing trend driven by local sources. Could you please look in more detail for this?*

As observed in the Section of Figure 7 from Grossi et al., 2016 presented at the beginning of this document the winter months at DEC are characterized by strong wind coming from the continental area of Spain which are reach in radon. This may explain the high concentration observed at the station during this period. It was not a single event but it has a repeatability over the different December months over the 7 years dataset. However, this fact needs to be further investigated, and it will be noted in the conclusions of the manuscript.

**L422-427:** *"In the present work, the correlation between observed wind and modelled wind (0.52) is higher than the correlation between observed and modelled radon concentrations (0.38 – 0.43). Moreover, differences between the three radon exhalation models are much lower than between observations and models. Therefore, although no observational data was available on BLH for DEC station, it can be deduced that most of the disagreement between models and observations may have come from the nocturnal boundary layer simulation rather than by radon exhalation maps uncertainties."* – *Could you clarify how you reached this conclusion? I'd like to understand the reasoning behind attributing most discrepancies to boundary layer simulations instead of the radon exhalation models. What specific evidence or analysis supports this interpretation? I'm not entirely convinced that relying solely on correlation coefficients provides a complete picture.*

Authors apologize for the lack of clearness of this paragraph. Actually, although it is clear that transport models have an important contribution to the discrepancies between model and observations, we can not determine the percentage of this contribution as we do not know the error of the radon flux maps. The phrase has been eliminated and a new phrase has been added:

*"However, overall differences between model and observations may be attributable to both radon flux maps and transport models and from the data obtained it is not possible to attribute a higher contribution in the uncertainties to the radon maps or to the atmospheric models."*

*L505-512: The authors compare the RTM results from 2013 to 2019, focusing on nocturnal accumulation, with the findings from Martinez-Eixarch et al. (2018), who conducted a campaign in 2015 using static chambers for approximately 1 to 3 days (10 am to 3 pm) each month across 15 spatially distributed locations.*

*It would be particularly insightful to include direct comparisons for the specific year of 2015. This year serves as a common reference point for both datasets, allowing for a more precise evaluation of how the RTM monthly averages align with the static chamber measurements from that same year. Highlighting these comparisons could clarify the performance of the RTM in accurately reflecting radon dynamics during 2015.*

The reviewer suggestion is really nice ad authors agree with it. Unfortunately, due to the datset holes, not enough data are available for the 2015 year for a specific comparison of the results with Martinez-Eixarch et al. (2018). It has to be specified that they found really low variability within years (2015 and 2016) in the methane fluxes.

References:

Cerralbo, P., Grifoll, M., Moré, J., Bravo, M., Sairouní Afif, A. and Espino, M.: Wind variability in a coastal area (Alfacs Bay, Ebro River delta), Adv. Sci. Res., 12(1), 11–21, doi:10.5194/asr-12-11-2015, 2015.

Grossi, C., Àgueda, A., Vogel, F. R., Vargas, A., Zimnoch, M., Wach, P., Martín, J. E., López-Coto, I., Bolívar, J. P., Morguí, J. A. and Rodó, X.: Analysis of ground-based 222Rn measurements over Spain: Filling the gap in southwestern Europe, J. Geophys. Res. Atmos., 121(18), 11,021-11,037, doi:10.1002/2016JD025196, 2016.

Grossi, C., Vogel, F. R., Curcoll, R., Àgueda, A., Vargas, A., Rodó, X., Morguí, J.-A., Grossi, C., Vogel, F. R. and By, C. C.: Study of the daily and seasonal atmospheric CH4 mixing ratio variability in a rural Spanish region using 222Rn tracer, Atmos. Chem. Phys., 18(8), 5847–5860, doi:10.5194/acp-18-5847-2018, 2018.

Hammer, S. and Levin, I.: Seasonal variation of the molecular hydrogen uptake by soils inferred from continuous atmospheric observations in Heidelberg, southwest Germany, Tellus B Chem. Phys. Meteorol., 61(3), 556–565, doi:10.1111/j.1600-0889.2009.00417.x, 2009.

Kozak, J. A., Reeves, H. W. and Lewis, B. A.: Modeling radium and radon transport through soil and vegetation, J. Contam. Hydrol., 66(3–4), 179–200, doi:10.1016/S0169-7722(03)00032-9, 2003.

Levin, I., Glatzel-Mattheier, H., Marik, T., Cuntz, M., Schmidt, M. and Worthy, D. E.: Verification of German methane emission inventories and their recent changes based on atmospheric observations, J. Geophys. Res. Atmos., 104(D3), 3447–3456, doi:10.1029/1998JD100064, 1999.

Levin, I., Karstens, U., Hammer, S., DellaColetta, J., Maier, F. and Gachkivskyi, M.: Limitations of the Radon Tracer Method (RTM) to estimate regional Greenhouse Gases (GHG) emissions – a case study for methane in Heidelberg, Atmos. Chem. Phys., 21(23), 1–34, doi:10.5194/acp-21-17907-2021, 2021.

Martínez-Eixarch, M., Alcaraz, C., Viñas, M., Noguerol, J., Aranda, X., Prenafeta-Boldu, F. X., Saldaña-De la Vega, J. A., del Mar Catala, M. and Ibáñez, C.: Neglecting the fallow season can significantly underestimate annual methane emissions in Mediterranean rice fields, PLoS One, 13(5), doi:10.1371/journal.pone.0198081, 2018.

Martínez-Eixarch, M., Alcaraz, C., Viñas, M., Noguerol, J., Aranda, X., Prenafeta-Boldú, F. X., Català-Forner, M., Fennessy, M. S. and Ibáñez, C.: The main drivers of methane emissions differ in the growing

and flooded fallow seasons in Mediterranean rice fields, Plant Soil, 460(1–2), 211–227, doi:10.1007/s11104-020-04809-5, 2021.

Megonigal, J. P., Brewer, P. E. and Knee, K. L.: Radon as a natural tracer of gas transport through trees, New Phytol., 225(4), 1470–1475, doi:10.1111/nph.16292, 2020.

Takeyama, Y., Ohsawa, T., Kozai, K., Hasager, C. B. and Badger, M.: Effectiveness of WRF wind direction for retrieving coastal sea surface wind from synthetic aperture radar, Wind Energy, 16(6), 865–878, doi:10.1002/we.1526, 2013.

Wada, A., Matsueda, H., Murayama, S., Taguchi, S., Hirao, S., Yamazawa, H., Moriizumi, J., Tsuboi, K., Niwa, Y. and Sawa, Y.: Quantification of emission estimates of CO2, CH4 and CO for east asia derived from atmospheric radon-222 measurements over the western North Pacific, Tellus, Ser. B Chem. Phys. Meteorol., 65(1), 1–16, doi:10.3402/tellusb.v65i0.18037, 2013.

---

## Referee Report (RR1)

Second review of "**Estimation of seasonal methane fluxes over a Mediterranean rice paddy area using the Radon Tracer Method (RTM)**" by Curcoll et al.

The authors have clearly put a lot of effort into this study, and revision of the original manuscript, which is to be commended given the significance of needing to characterise changing methane missions related to agricultural practices and other anthropogenic activities. The primary goal of the study is to characterise the seasonality in methane emission specifically from the rice paddy growing area of the Ebro River Delta region. After revision however, I still have reservations that the RTM is an appropriate method to achieve the intended goals. I outline my main concerns below.

Application of the RTM to this particular scenario challenges many of the fundamental assumptions on which the technique is based (some of which are reviewed by the authors in Section 2.3 of the revised MS). The fluxes of radon and $CH_4$ over the contributing footprint should ideally be relatively consistent and similarly distributed (or, if not, randomly correlated on small spatial scales of heterogeneity, Levin et al 2021), the radon flux in particular should be well characterised for the period of measurement, accumulation amounts over the nocturnal window should well exceed instrument uncertainty, and for such a restricted study region, strictly stable nocturnal conditions should be adhered to.

The radon tracer method provides a footprint-weighted average estimate of (in this case) the $CH_4$ flux. Of the estimated contributing footprint region (Fig 10), the authors indicate that 50% constitutes the ERD (~65% of which are rice fields), 35% is ocean and 15% are other regions of NE Spain (with relative contributions from each of these regions changing seasonally). Each of these 4 contributing regions are characterised by substantially different radon and $CH_4$ emission characteristics (roughly speaking, according to the MS and supp material provided):

**NE Spain:**      moderate to high Rn flux, moderate $CH_4$ flux (~**15%**).
**Ocean:**         almost zero radon flux (~0.2 mBq m$^{-3}$), assumed zero $CH_4$ flux (~**35%**).
**Non-rice ERD:**  low radon flux, low to intermediate $CH_4$ flux (~**20%**).
**Rice fields:**   seasonally varying radon flux from near zero to moderate, seasonally varying $CH_4$ flux from low to high (~**30%**).

Even if the oceanic contribution can be ignored as the authors suggest, the rice field region could represent slightly less than half of the contributing fetch region in the reported results.

Contributions to $CH_4$ emissions from the non-rice paddy regions may well be small (as the authors suggest), but I suspect not entirely negligible. Given the prevalence of high winds in the region (which will "flatten" diurnal cycles), it would have been informative to see composite diurnal cycles (e.g. Fig 5) for just the days of each month where the RTM had been applied, to better assess this assertion. Also, having half of the non-oceanic contributing fetch region for the reported results characterised by a smaller $CH_4$ flux, wouldn't this result in an underestimation of the actual $CH_4$ flux coming from the rice fields? After all, the estimated flux from the RTM is assumed to have been uniformly contributing to concentrations within the "idealised box" of the model for each hour of the accumulation window.

Equation (3) of the manuscript shows how (for similarly distributed fluxes), the flux of $CH_4$ can be retrieved by scaling the slope of the co-measured trace gases with the radon flux. It also highlights that any uncertainty or error in the estimate of the radon flux is directly proportional to subsequent error / uncertainty in the $CH_4$ flux. The radon flux estimates for this study, derived from the Karstens and Levin (2023) flux map with different soil moisture reanalyses, account for typical seasonal variability of soil moisture within the footprint weighted fetch region, but do not account for

saturation (or worse, complete inundation). In most studies, saturation alone is considered to severely inhibit (or stop) radon emission (e.g., Griffiths et al 2010, doi:10.5194/acp-10-8969-2010). Information provided in this MS indicates that the rice fields could be flooded for around 7 months of the year. Radon fluxes from open water bodies such as oceans are near zero (~0.2 mBq m$^{-3}$; e.g., Zahorowski et al. 2013, http://dx.doi.org/10.3402/tellusb.v65i0.19622). For conditions of complete inundation (8 – 15 cm deep), since the water is shallow compared with an ocean, and a limited amount of radon transfer through crop stems may be possible, a slightly larger radon flux may be expected. Even assuming a factor of 4 increase in radon flux for inundation conditions (i.e., 0.8 mBq m$^{-3}$), this flux is still an order of magnitude less than the flux assumed in this study for the target area. Assuming that the derived radon fluxes derived for the non-inundation periods are representative (which I believe they are), if the flux estimates presented were actually predominantly representative of only the rice-paddy region (L447-448 revised manuscript), the $CH_4$ flux estimates derived in this study for the inundated periods would necessarily be ~10 times higher than they actually are, on account of the overestimated radon flux.

Another assumption of the RTM is that the fluxes of radon and the tracer species are similarly distributed – here, for 7 months of the year – there is a high $CH_4$ flux from the primary region of interest, and a radon flux close to zero. This is another challenge for the suitability of the technique for this purpose, and the claimed specificity of the results.

Some further concerns regarding the fetch/flux distribution over the study region. A fundamental assumption of the simplified model used to derive the RTM (Equation 3) is that observed concentrations within the "idealised accumulation volume" over the accumulation window are a function ONLY of the surface flux and atmospheric mixing depth. Looking at a picture of the site location on the Ebro River Delta peninsula (Figure S1), north and east of the site there is rice paddy fetch for about 2 km, south of the site there is around 8 km of rice paddy fetch, and west of the site, approximately 15-18 km of rice paddy fetch. Beyond these bounds the authors consider $CH_4$ fluxes to be zero or close to negligible. Bearing in mind that according to L511, the decision was made NOT to restrict nocturnal wind speeds to below 1.5 m s$^{-1}$ for the final results, even at a best-case scenario of 1.5 m s$^{-1}$, air would move from the ocean to the site (from the north or east) in 30 minutes (about 35% of the time according to Fig 10). From the south, air masses would transit the rice-paddy fields in 90 minutes, and from the west (after passing over interior Spain), the air masses would transit the rice-paddy fields in 3 hours. Given that the chosen nocturnal accumulation period for this study was 6 hours, it is quite clear that the amount of $CH_4$ accumulating in the "idealised box" will also be a strong function of wind direction (violating a basic assumption of the method), and that rice-paddy $CH_4$ sources would influence observed concentrations for typically less than half of the chosen accumulation window. It is therefore quite likely that a large proportion of the observed positive correlations between radon and $CH_4$ during selected RTM events were either due to influences on the air mass that happened much further upstream of the ERD or based on $CH_4$-to-Rn slopes derived from only 2 or 3 measurement points.

The linearity of the $CH_4$-to-Rn slope (on which equation 3 is based) reflects the degree to which the source functions of the gases in question are similarly distributed (a requirement of the technique). The diurnal cycles of $CH_4$ and radon shown in Fig 5 indicate substantially different accumulation behaviour between the two gases in this study – presumably related to substantial difference in source distributions. The majority of contemporary RTM studies adopt a linearity threshold of $0.6 \leq R^2 \leq 0.8$ in an attempt to minimise contributions to results from conditions that don't meet the assumptions of the technique. In this study an $R^2$ threshold of only 0.5 was adopted, which has implications for the derived flux uncertainties.

Another contributing factor to the low $R^2$ threshold adopted in this study is likely to have been the signal-to-noise ratio for the ARMON's radon detection in this coastal and often water-logged environment. Fig 5 of the revised MS indicates typical whole-night radon accumulation of 1.5 – 3 Bq m$^{-3}$. On the more stable nights, Fig 8 indicates maximum nocturnal radon accumulations over a whole night of 2 – 4 Bq m$^{-3}$ (less for the 6-hour accumulation window used for this study). As part of the recently completed EMPIR 19ENV01 traceRadon Project, the performance of an *improved* model of ARMON (relative to the one used in this study) was tested over radon concentrations between 0 – 40 Bq m$^{-3}$ under controlled conditions. In this idealised case, the standard deviation of hourly measurements was ~3 Bq m$^{-3}$ (see below, from Röttger et al. 2024; XIV IMEKO World Congress "Think Metrology", 26-29 Aug, Hamburg Germany), which is large compared with hourly radon accumulation rates of 0.3 – 0.6 Bq m$^{-3}$. This type of instrument would be better suited to sites with larger nocturnal radon accumulation ranges (e.g., Grossi et al. 2018, https://doi.org/10.5194/acp-18-5847-2018).

[Figure]

*Figure 4: Statistical spread of the difference between measured values and the reference values derived from the sources as histogram in linear presentation. The histograms are normalised to equal surface. The numbers given are the respective standard deviations of the histogram in Bq·m$^{-3}$. The coloured area is the 95 % coverage interval for the respective histogram.*

Including nights for RTM calculations on which positive correlations between radon and CH$_4$ are observed, but wind speeds are significantly above 1.5 m s$^{-1}$, greatly increases the risk of incorporating flux contributions from (i) large distances upstream (not representative of the study region), or (ii) synoptic scale fetch change influences (which violate important assumptions of the RTM).

Separate to their derivation of a contributing footprint region for this study, the authors have equated a measurement height of 10 m agl with observing only local influences (e.g., L447-448 of revised manuscript). Certainly, under convective daytime conditions, the dominant measurement fetch is usually within <100 times the measurement height. Under stable nocturnal conditions however, a requirement for the application of the "local" implementation of the RTM, as described by Conen et al (2002; GRL, 10.1029/2001GL013429), a column of air will move across the surface at the average near surface wind speed, accumulating influences of surface fluxes over which it passes. At wind speeds of only 1.5 m s$^{-1}$, over a 6-hour nocturnal accumulation window, the observed air mass will have been influenced by a fetch of at least 32 km. Over this distance, at this study site, air masses could easily pass over 4 regions with entirely different surface emission characteristics. As previously mentioned, a fundamental assumption of the nocturnal accumulation implementation of the RTM is that over the accumulation window concentrations within the "idealised atmospheric column" are purely a function of a near-constant source of both gases (per night) and a changing mixing depth.

For all the reasons above I am not convinced that Fig 12 represents the true seasonal variability of $CH_4$ fluxes specifically from the rice-paddy region of the ERD.

Scott Chambers, Research Scientist, ANSTO
31/1/2025

---

## Author Response (AR2)

**egusphere-2024-1370**
**Title: Estimation of seasonal methane fluxes over a Mediterranean rice paddy area using the Radon Tracer Method (RTM)**
**Author(s): Roger Curcoll et al.**

**Comments to the second review by Scott Chambers**

The authors have clearly put a lot of effort into this study, and revision of the original manuscript, which is to be commended given the significance of needing to characterize changing methane missions related to agricultural practices and other anthropogenic activities. The primary goal of the study is to characterize the seasonality in methane emission specifically from the rice paddy growing area of the Ebro River Delta region. After revision however, I still have reservations that the RTM is an appropriate method to achieve the intended goals. I outline my main concerns below.

Application of the RTM to this particular scenario challenges many of the fundamental assumptions on which the technique is based (some of which are reviewed by the authors in Section 2.3 of the revised MS). The fluxes of radon and CH4 over the contributing footprint should ideally be relatively consistent and similarly distributed (or, if not, randomly correlated on small spatial scales of heterogeneity, Levin et al 2021), the radon flux in particular should be well characterized for the period of measurement, accumulation amounts over the nocturnal window should well exceed instrument uncertainty, and for such a restricted study region, strictly stable nocturnal conditions should be adhered to.

First of all, the authors want to thank the third reviewer for his accurate work and his time. However, the authors think that most of the concerns of the reviewer were already addressed during the first round of this revision process. Considering that the other two reviewers already accepted the document after the first round, here authors make an extra effort to clarify the different points to the third reviewer.

As it was already explained during the first round of revisions, an harmonized protocol for the application of the Radon Tracer Method is not yet available for the scientific community and studies like the present one are essential to understand its limitations and strengths for better addressing future improvements.

For the preparation of this document the authors benefited from three recent publications (two of which are currently under revision) that explore themes related to RTM application, the reliability of radon flux models, and a comparison of the ARMON and ANSTO 200L instruments. These publications include: Yver-Kwok et al (https://egusphere.copernicus.org/preprints/2024/egusphere-2024-3107/); Maier et al. (https://egusphere.copernicus.org/preprints/2025/egusphere-2025-477/); Röttger et al. (https://www.sciencedirect.com/science/article/pii/S2665917424006846). This is important not only for demonstrating that the methodology of the present manuscript is supported by existing literature, but also for emphasizing the ongoing discussions in this research field and the importance of this study for future developments.

It is true that a homogeneous distribution of radon and target gas fluxes over the area of interest is an important factor for applying the RTM. However, in a region such as the Ebro River Delta (where there are essentially no other significant target gas emissions besides those under study, primarily due to the high methane emissions from rice paddies) the analysis of variability and the application of the RTM become significantly more straightforward.
The Ebro River Delta, with a surface of 320 km$^2$, cannot be considered a point source, also taking into account the low height of the station tower (only 10 meters above the ground).

Finally, as suggested by both reviewers in the first round of revision, stable nocturnal conditions were taken into account by filtering the study to include only low wind speed events.

So far, based on RTM assumptions, past studies were conducted primarily in relatively 'simple' areas of Central Europe, where radon and target gas concentrations are lower and quite homogeneous over time and space. The present study, however, is particularly challenging as it applies the RTM in an 'extreme' region (i.e. one where radon flux models and atmospheric transport models do not operate as smoothly). Such regions exist across Europe, especially in Mediterranean and Oceanic stations in the south, and emissions must also be quantified there.

Despite the challenges, the results obtained here are remarkable, as noted by both reviewers and the editor. The agreement between RTM-based estimates and manual accumulation chamber measurements (which are time- and cost-intensive) is striking. For instance, in October RTM-based results estimated a median flux of 13.9 mg $CH_4$ $m^{-2}$ $h^{-1}$, closely matching fluxes from static chambers, which were calculated as 14.7 ± 4.2 mg $CH_4$ $m^{-2}$ $h^{-1}$.

The average difference between RTM-based $CH_4$ fluxes obtained using the Flex-WRF-ERA5 and Flex-WRF-GLDAS models was only 0.1 ± 0.2 mg $CH_4$ $m^{-2}$ $h^{-1}$, and this difference remained stable regardless of whether the wind threshold of 1.5 m $s^{-1}$ was applied.

This study shows the great potential that RTM offers for emission budget analysis, with the theoretical, modeling and experimental efforts done by authors contributing to robust and reliable results. While uncertainties exist and must be acknowledged, they provide valuable insights for future research in this field.

A key finding of this study is that the EDGAR v7.0 inventory, which accounts for emissions from agricultural soils, does not consider rice paddy field emissions at ERD. This is evident form the fact that the agricultural soil emissions assigned to ERD pixels are reported as below 0.02 mg $CH_4$ $m^{-2}$ $h^{-1}$.

The radon tracer method provides a footprint-weighted average estimate of (in this case) the $CH_4$ flux. Of the estimated contributing footprint region (Fig 10), the authors indicate that 50% constitutes the ERD (~65% of which are rice fields), 35% is ocean and 15% are other regions of NE Spain (with relative contributions from each of these regions changing seasonally). Each of these 4 contributing regions are characterised by substantially different radon and $CH_4$ emission characteristics (roughly speaking, according to the MS and supp material provided):
NE Spain: moderate to high Rn flux, moderate $CH_4$ flux (~15%).
Ocean:almost zero radon flux (~0.2 mBq m-3), assumed zero $CH_4$ flux (~35%).
Non-rice ERD:low radon flux, low to intermediate $CH_4$ flux (~20%).
Rice fields: seasonally varying radon flux from near zero to moderate, seasonally varying $CH_4$ flux from low to high (~30%).
Even if the oceanic contribution can be ignored as the authors suggest, the rice field region could represent slightly less than half of the contributing fetch region in the reported results. Contributions to CH4 emissions from the non-rice paddy regions may well be small (as the authors suggest), but I suspect not entirely negligible. Given the prevalence of high winds in the region (which will "flaten" diurnal cycles), it would have been informative to see composite diurnal cycles (e.g. Fig 5) for just the days of each month where the RTM had been applied, to better assess this assertion. Also, having half of the non-oceanic contributing fetch region for the reported results characterised by a smaller CH4 flux, wouldn't this result in an underestimation of the actual CH4 flux coming from the rice fields? After all, the estimated flux from the RTM is assumed to have been uniformly contributing to concentrations within the "idealised box" of the model for each hour of the accumulation window.

The authors thank the reviewer for this comment. Before addressing the response, we would like to clarify that in this study radon and methane concentrations measured at the station are used within the RTM, together with the underline{effective} radon flux observed at the station. The effective radon flux is derived from the weighted contribution of initial radon flux maps considering the residence time of FLEXPART back trajectories.

Regarding the potential presence of other methane sources within the station footprint, our analysis, which is consistent with the EDGAR emission inventories, Corine Land Cover map, and other bibliographic studies, indicates that their contributions are not significant compared to emissions from rice paddies. Specifically, methane fluxes over the footprint area, when rice paddies are dry, range between 0.3 and 0.6 mg $CH_4$ $m^{-2}$ $h^{-1}$, whereas our results, corroborated by literature (Martínez-Eixarch et al., 2018) and other studies in rice fields (Wang et al., 2018), show that methane fluxes from rice fields exceed 3 mg $CH_4$ $m^{-2}$ $h^{-1}$ between July and November. Considering this difference, the external contribution to the rice paddies methane emissions can be considered uncertain within a range of 10 % to 15 %.

As suggested by Dr. Chambers, the following figures show diurnal cycles of radon and methane concentrations for each month considering only the days when RTM was applied. The first figure presents the RTM dataset with a wind speed threshold of 1.5 m/s, and the second figure shows the dataset without this threshold. No important differences are observed between these figures and the results obtained using the complete dataset (Figure 5 of the manuscript), except for February, where deviations are due to the limited amount of available data.

[Figure]

*Radon and methane composite diurnal cycles for just the days of each month where the RTM was applied and with a wind speed threshold of 1.5 m/s.*

[Figure]

*Radon and methane composite diurnal cycles for just the days of each month where the RTM was applied and with no wind speed threshold.*

Regarding the footprint area, the reviewer may consider that nocturnal winds may be overestimated by the atmospheric transport model based on WRF (see Figure 7), which may, in turn, lead to an overestimation of the footprint area. Moreover, it is well known that the WRF model does not accurately reproduce the height of the nocturnal boundary layer. As for methane fluxes, as the reviewer says, they may indeed be underestimated because the footprint region is larger than the Ebro Delta. However, as clarified in the methodology section of the manuscript, the use of the FLEXPART transport model allowed the calculation of the footprint area for each selected RTM night.

Equation (3) of the manuscript shows how (for similarly distributed fluxes), the flux of $CH_4$ can be retrieved by scaling the slope of the co-measured trace gases with the radon flux. It also highlights that any uncertainty or error in the estimate of the radon flux is directly proportional to subsequent error / uncertainty in the CH4 flux. The radon flux estimates for this study, derived from the Karstens and Levin (2023) flux map with different soil moisture reanalyses, account for typical seasonal variability of soil moisture within the footprint weighted fetch region, but do not account for saturation (or worse, complete inundation). In most studies, saturation alone is considered to severely inhibit (or stop) radon emission (e.g., Griffiths et al 2010, doi:10.5194/acp-10-8969-2010).
Information provided in this MS indicates that the rice fields could be flooded for around 7 months of the year. Radon fluxes from open water bodies such as oceans are near zero (~0.2 mBq m-3; e.g.,

Zahorowski et al. 2013, htp://dx.doi.org/10.3402/tellusb.v65i0.19622). For conditions of complete inundation (8 – 15 cm deep), since the water is shallow compared with an ocean, and a limited amount of radon transfer through crop stems may be possible, a slightly larger radon flux may be expected. Even assuming a factor of 4 increase in radon flux for inundation conditions (i.e., 0.8 mBqm-3), this flux is still an order of magnitude less than the flux assumed in this study for the target area. Assuming that the derived radon fluxes derived for the non-inundation periods are representative (which I believe they are), if the flux estimates presented were actually predominantly representative of only the rice-paddy region (L447-448 revised manuscript), the CH4 flux estimates derived in this study for the inundated periods would necessarily be ~10 times higher than they actually are, on account of the overestimated radon flux.

Thanks again to the reviewer for this comment. Before addressing the response, we would like to emphasize that the radon flux used in Equation 3 is calculated for each RTM night using radon flux maps (Karstens and Levin, 2023) and weighting them according to the residence time of air masses arriving at DEC station. This means that the radon and methane concentrations measured at the station each night are due to the contribution not only from rice fields but also from other regions where air masses have been in contact with the surface gas emissions.

The key point here is that, in the case of radon, the difference between dry and water-saturated areas within the ERD is relatively small, whereas methane emissions from rice fields during straw incorporation are significantly higher in comparison with other sources. The strong agreement between our results and those obtained by Martínez-Eixarch et al. (2018) using an independent methodology support the reliability of our study.

Additionally, we acknowledge that uncertainty in traceRadon radon flux maps is high and may be further improved as reported by Maier et al. (2025). Although radon maps have seen significant improvements over the last decades, there are still lots of uncertainties. For example, a recent study on the uncertainty of radon fluxes obtained from the two available maps (ERA5-Land and GLAS-Noah) showed that, in central Europe, fluxes estimated with GLDAS-Noah could be twice as high as those estimated using ERA5 during certain periods (see Curcoll's PhD dissertation (2024) and Maier et al. (2025) work).

The authors are fully aware of the large uncertainties associated with these results. A new section, reproduced here below, has been added in the manuscript in order to assess the uncertainty associated in this work. We also recognize the importance of carrying out new studies to quantify and minimize these uncertainties.

**3.5 Uncertainty and representativeness of the RTM-based CH₄ fluxes at DEC**

One of the simplifications considered for the application of the RTM is that fluxes should be homogeneous around the station, as strong point sources could affect the concentrations measured at the station and the regional flux estimation. Although the ERD is relatively small compared to other areas where the RTM has been applied (Schmidt et al. 1996, Levin et al. 2021), the short height of the sampling tower and the homogeneity of the ERD makes this area well-suited for the application of the RTM.

The difficulty of the application of the RTM in this study is sure but the evaluation of both meteorological and atmospheric transport models for the area and periods of interest, the comparison of the RTM based methane fluxes results with national inventories and other independent experimental studies from the bibliography add useful highlights for the research in this field and improvement of the RTM application protocols.

As seen in section 3.2, the WRF model in regions like ERD does not always simulate correctly the nocturnal accumulation and the wind speeds, deriving to important bias in concentration simulations. The advantage of the work exposed here is that due to the short height of the sampling point, the footprint of the station is quite small, within a few kilometers, and thus, the footprint is more reliable. Moreover, by applying the 1.5 m/s wind speed threshold for selecting RTM-feasible night events, we ensure minimal advection from continental sources. In agreement with EDGAR emission inventories, Corine Land Cover map, and other bibliographic studies, there are no significant contributions to the methane emissions compared to those from rice paddies. Methane fluxes over the footprint area and when rice paddies are dried are between 0.3 and 0.6 mg CH4 $m^{-2}$ $h^{-1}$, while our results, corroborated by literature (Martínez-Eixarch et al., 2018) and other studies in rice fields (Wang et al., 2018), show methane fluxes due to the rice fields above 3 mg $CH_4$ $m^{-2}$ $h^{-1}$ between July and November. Considering these differences, the external contribution to the rice paddies methane emissions may be assumed with an uncertainty between 10-15%.

One of the elements that produces more uncertainty in the calculus of the methane flux using the RTM is the uncertainty of real radon flux distribution over the footprint area of the measurement station (Vogel et al., 2012, Levin et al., 2021). In Eq. (3), the radon flux is directly proportional to the estimated methane flux. Therefore, an error in the radon flux will proportionally produce an error in the estimated methane flux. For this reason, until the radon flux maps in the area over the DEC station could be sufficiently validated, the global annual methane flux estimated with the RTM should be carefully accepted assuming a certain uncertainty. On the other hand, the integration, in this work, of data from several years, makes the result more robust than if using data from a single year.

One of the limitations of the RTM is that only the nocturnal emissions are monitored. In the case of rice fields, it is well known that the gross ecosystem photosynthesis (GEP) and the soil temperature are drivers of $CH_4$ flux variability (Hatala et al., 2012). Although diel fluxes and nocturnal fluxes keep a strong correlation (Wassmann et al., 2018), methane emissions in the early afternoon can be between 10% and 20% higher than the nocturnal emissions during the productive months (Alberto et al., 2014; Dai et al., 2019; Minamikawa et al., 2012). This difference may lead to an underestimation, ranging between 10% (Weller et al., 2015) and 20 % (Wassmann et al., 2018), of diel fluxes if considering only the nocturnal emissions.

Another assumption of the RTM is that the fluxes of radon and the tracer species are similarly distributed – here, for 7 months of the year – there is a high CH4 flux from the primary region of interest, and a radon flux close to zero. This is another challenge for the suitability of the technique for this purpose, and the claimed specificity of the results.

The ERD area has homogeneously distributed emissions. As already mentioned, based on RTM assumptions, previous studies obtained results in 'simple' areas of central Europe where radon and target gas concentrations are lower and quite homogeneous over time and space. Our study was challenging because, for the first time, we applied RTM in a 'extreme' region and, despite the challenge, the results were stunning.

Some further concerns regarding the fetch/flux distribution over the study region. A fundamental assumption of the simplified model used to derive the RTM (Equation 3) is that observed concentrations within the "idealised accumulation volume" over the accumulation window are a function ONLY of the surface flux and atmospheric mixing depth. Looking at a picture of the site location on the Ebro River Delta peninsula (Figure S1), north and east of the site there is rice paddy fetch for about 2 km, south of the site there is around 8 km of rice paddy fetch, and west of the site,

approximately 15-18 km of rice paddy fetch. Beyond these bounds the authors consider CH4 fluxes to be zero or close to negligible. Bearing in mind that according to L511, the decision was made NOT to restrict nocturnal wind speeds to below 1.5 m s-1 for the final results, even at a best-case scenario of 1.5 m s-1, air would move from the ocean to the site (from the north or east) in 30 minutes (about 35% of the time according to Fig 10). From the south, air masses would transit the rice-paddy fields in 90 minutes, and from the west (atier passing over interior Spain), the air masses would transit the rice-paddy fields in 3 hours. Given that the chosen nocturnal accumulation period for this study was 6 hours, it is quite clear that the amount of CH4 accumulating in the "idealised box" will also be a strong function of wind direction (violating a basic assumption of the method), and that rice-paddy CH4 sources would influence observed concentrations for typically less than half of the chosen accumulation window. It is therefore quite likely that a large proportion of the observed positive correlations between radon and CH4 during selected RTM events were either due to influences on the air mass that happened much further upstream of the ERD or based on CH4-to-Rn slopes derived from only 2 or 3 measurement points.

Authors are not really sure to understand which is the point of the discussion here. As the reviewer may see in the first round of comments, the authors DID apply the threshold criteria of low wind speeds, and the comparison of the results shows basically no differences between the full dataset and low winds dataset. On this basis, the decision of carrying on the analysis with the full dataset was to ensure a robust statistic for the results. The correlation criteria, on the other hand, was applied to only select nights where the increase of the radon and methane concentrations was proportional.

The linearity of the CH4-to-Rn slope (on which equation 3 is based) reflects the degree to which the source functions of the gases in question are similarly distributed (a requirement of the technique). The diurnal cycles of CH4 and radon shown in Fig 5 indicate substantially different accumulation behaviour between the two gases in this study – presumably related to substantial difference in source distributions. The majority of contemporary RTM studies adopt a linearity threshold of $0.6 \leq R2 \leq 0.8$ in an atempt to minimise contributions to results from conditions that don't meet the assumptions of the technique. In this study an R2 threshold of only 0.5 was adopted, which has implications for the derived flux uncertainties.

Thanks to the reviewer for this comment but it was already clarified during the first round of comments that the use of an $R^2 \geq 0.5$ was already been acknowledged in past scientific studies. Authors agree on the fact that a higher $R^2$ would give more robust results. On the contrary it will also give a smaller dataset which will lead to higher statistical uncertainty. The compromise between robust statistics and reliable data was here considered a priority.

Another contributing factor to the low R2 threshold adopted in this study is likely to have been the signal-to-noise ratio for the ARMON's radon detection in this coastal and otien water-logged environment. Fig 5 of the revised MS indicates typical whole-night radon accumulation of 1.5 – 3 Bq m-3. On the more stable nights, Fig 8 indicates maximum nocturnal radon accumulations over a whole night of 2 – 4 Bq m-3 (less for the 6-hour accumulation window used for this study). As part of the recently completed EMPIR 19ENV01 traceRadon Project, the performance of an improved model of ARMON (relative to the one used in this study) was tested over radon concentrations between 0 – 40 Bq m-3 under controlled conditions. In this idealised case, the standard deviation of hourly measurements was ~3 Bq m-3 (see below, from Rötger et al. 2024; XIV IMEKO World Congress "Think Metrology", 26-29 Aug, Hamburg Germany), which is large compared with hourly radon accumulation rates of 0.3 – 0.6 Bq m-3. This type of instrument would be beter suited to sites with larger nocturnal radon accumulation ranges (e.g., Grossi et al. 2018, htps://doi.org/10.5194/acp-18-5847-2018).

Thanks to the reviewer for this comment. Looking at the bibliography we can see that Figure 2b from Grossi et al. (2020) shows an intercomparison between the ARMON instrument (red circles), used in the present study, and other radon and radon progeny monitors. One of them is the ANSTO monitor (blue circles) provided and sold in Europe by Dr Chamber, the reviewer of the present manuscript. This figure clearly shows the ability of the ARMON to measure increments of concentration lower than 3 Bq m$^{-3}$.

[Figure]

**Figure 2. (a)** Hourly time series of the atmospheric $^{222}$Rn and, in the case of LSCE and HRM data, $^{214}$Po activity concentration, measured at the Orme de Mérisiers (ODM) station during Phase I (between 25 November 2016 and 23 January 2017) by the ARMON (red circles), ANSTO_ODM (blue circles), HRM (green circles), and LSCE (orange circles) monitors. **(b)** Hourly time series of the atmospheric $^{222}$Rn and $^{214}$Po measured between 27 December 2016 and 4 January 2017.

In a study published by Curcoll et al. (2024), the authors showed that for a typical radon concentration of 5 Bq m$^{-3}$ the total uncertainty of the ARMON instrument was about 10% (k=1), that is an uncertainty of about 0.5 Bq m$^{-3}$. This result confirms that the instrument (nowadays commercialized by Radonova, a leading company in radon measurements) is able to capture radon concentrations increases similar to the ones observed in the present study at the DEC station during the nocturnal accumulation periods. In the case of the ANSTO, for example, Maier et al. (2025) shows that using this instrument may lead to an overestimation of the radon concentration due to its intrinsic background. This means that a full metrology is needed (and is under construction) for atmospheric radon measurements in Europe.

Finally, in the same study suggested by the reviewer (Röttger et al., 2025), it can be observed that the ARMON instrument has a high linear response with a very low standard deviation of the scatter (0,77%), which is much better than the one from other commercially available radon and radon progeny atmospheric monitors. Röttger et al. (2025) also underlines the very quick response of the ARMON monitor to a concentration pulse. This makes the instrument quite suitable to measure small concentrations variations over short time periods. Below you can see the figures from the cited manuscript.

[Figure]

Figure 6: Results for the calibration factor of the ARMON v2 detector applying all sources and combinations measured, taking correlated and uncorrelated contributions of the source characterisation into account.

[Figure]

Figure 7: 3-dimensional representation of the time and energy (channel) resolved response of the ARMON v2 detector for a defined $^{222}$Rn pulse applied at time 0 at the air inlet of the detector. At the lower channels (5-10) $^{218}$Po represents the faster response of the detector and channel (20-30) the slower response of $^{214}$Po is visible.

Including nights for RTM calculations on which positive correlations between radon and CH4 are observed, but wind speeds are significantly above 1.5 m s-1, greatly increases the risk of incorporating flux contributions from (i) large distances upstream (not representative of the study region), or (ii) synoptic scale fetch change influences (which violate important assumptions of the RTM).

Thanks to Dr Chambers for this comment. However, the authors already answered to this comment in the first revision and in this document. Again, they declare that winds above 1.5 m/s have been ONLY included for the calculation of the annual methane emission, and it has been done because the difference in methane flux between the two data series in the months with enough data were smaller than 15%.

Separate to their derivation of a contributing footprint region for this study, the authors have equated a measurement height of 10 m agl with observing only local influences (e.g., L447-448 of revised manuscript). Certainly, under convective daytime conditions, the dominant measurement fetch is usually within <100 times the measurement height. Under stable nocturnal conditions however, a requirement for the application of the "local" implementation of the RTM, as described by Conen et al (2002; GRL, 10.1029/2001GL013429), a column of air will move across the surface at the average near surface wind speed, accumulating influences of surface fluxes over which it passes.
At wind speeds of only 1.5 m s-1, over a 6-hour nocturnal accumulation window, the observed air mass will have been influenced by a fetch of at least 32 km. Over this distance, at this study site, air masses could easily pass over 4 regions with entirely different surface emission characteristics. As previously mentioned, a fundamental assumption of the nocturnal accumulation implementation of the RTM is that over the accumulation window concentrations within the "idealised atmospheric column" are purely a function of a near-constant source of both gases (per night) and a changing mixing depth.

As commented before, the use of back trajectories coupled with the radon flux maps allow to pinpoint the area of influence. The use of these back trajectories and the limitation of 1.5 m s⁻¹reduce the influence of the regions outside the ERD. The continental influence has been already estimated to be around 15%.

For all the reasons above I am not convinced that Fig 12 represents the true seasonal variability of CH4 fluxes specifically from the rice-paddy region of the ERD.

The authors of this study really regret that one of the three reviewers of the study has still concerns about the methodology and the results obtained and shown in this manuscript. The authors believe that the methodology of the RTM has been here adjusted and applied in the best way. Additionally, despite the uncertainties coming with the results (mainly due to the application of radon flux maps and atmospheric transport models in frontier region such as the delta of a Mediterranean river), the amazing agreement between the methane fluxes, their seasonal variability, as well as their absolute mean values, obtained in this study and those obtained by Martínez-Eixarch et al. (2018) using a completely independent method is stunning. This result, together with the comparison with official emission inventories, confirm the importance of this study and its publication. Research results are never perfect, but they are milestones to build and to improve knowledge.

[Figure]

485    **Figure 12. Boxplot of the methane RTM based fluxes over the DEC station calculated using Flex-WRF-ERA5 (blue) and Flex-WRF-GLDAS (red) for every month of the year (period 2013-2019), and methane fluxes over ERD using static chambers (green points**